



# Absorption enhancement of BC particles in a Mediterranean city and countryside: effect of PM chemistry, aging and trend analysis

Jesús Yus-Díez[1,2], Marta Via[1,2], Andrés Alastuey[1], Angeliki Karanasiou[1], María Cruz Minguillón[1], Noemí Perez[1], Xavier Querol[1], Cristina Reche[1], Matic Ivančič[3], Martin Rigler[3], and Marco Pandolfi[1]

[1]Institute of Environmental Assessment and Water Research (IDAEA-CSIC), Barcelona, 08034, Spain
[2]Grup de Meteorologia, Departament de Física Aplicada, Universitat de Barcelona, C/Martí i Franquès, 1, 08028, Barcelona, Spain
[3]Aerosol d.o.o., Ljubljana, Slovenia

**Correspondence:** Jesús Yus-Díez (jesus.yus@idaea.csic.es) and Marco Pandolfi (marco.pandolfi@idaea.csic.es)

**Abstract.**

Black carbon (BC) is recognized as the most important warming agent among atmospheric aerosol particles. The absorption efficiency of pure BC is rather well known, nevertheless the mixing of BC with other aerosol particles can enhance the BC light absorption efficiency, thus directly affecting the Earth radiative balance. The effects on climate of the BC absorption

enhancement due to the mixing with these aerosols is not yet well constrained because these effects depend on the availability of material for mixing with BC, thus creating regional variations.

Here we present the mass absorption cross-section, MAC, and absorption enhancement of BC particles, ($E_{abs}$), at different wavelengths (from 370 nm to 880 nm for on-line measurements and at 637 nm for off-line measurements) measured at two sites in the Western Mediterranean, namely Barcelona (BCN; urban background) and Montseny (MSY; regional background).

$E_{abs}$ values ranged between 1.24 and 1.51 at the urban station depending on the season and wavelength used as well as on the pure BC MAC used as a reference. The largest contribution to $E_{abs}$ was due to the internal mixing of BC particles with other aerosol compounds, on average between a 91 and a 100 % at 370 and 880 nm, respectively. Additionally, 14.5 and 4.6% of the total enhancement at the short-UV (370 nm) was due to externally mixed BrC particles during the cold and the warm period, respectively. On average, at MSY station, a higher $E_{abs}$ value was observed (1.83 at 637 nm) compared to BCN (1.37 at 637

nm), which was associated to the higher fraction of organic aerosols available for BC coating at the regional station, as denoted by the higher OC:EC ratio observed at MSY compared to BCN. At both BCN and MSY $E_{abs}$ showed an exponential increase with the amount of non-refractory (NR) material available for coating ($R_{NR-PM}$). The $E_{abs}$ at 637 nm at MSY regional station reached values up to 3 during episodes with high $R_{NR-PM}$, whereas in BCN $E_{abs}$ kept values lower than 2 due to the lower relative amount of coating materials measured at BCN compared to MSY. The main sources of organic aerosols influencing

$E_{abs}$ throughout the year were HOA and COA (primary OA from traffic and cooking emissions, respectively) at both 370 nm and 880 nm. At the short-UV wavelength (370 nm), a strong contribution to $E_{abs}$ from BBOA (biomass burning OA) and LO-OOA (less-oxygenated OA) sources was observed in the colder period. Moreover, we found an increase of $E_{abs}$ with the aging state of the particles, especially during the colder period. This increase of $E_{abs}$ with particle aging was associated to a larger relative amount of secondary organic aerosols (SOA) compared to primary OA (POA). The availability of a long dataset at both





stations from off-line measurements enabled a decade-long trend analysis of $E_{abs}$ at 637 nm, that showed positive statistically significant trends of $E_{abs}$ during the warmer months at MSY station. This s.s. positive trend at MSY mirrored the observed increase of the OC:EC ratio with time. Moreover, in BCN during the COVID-19 lockdown in spring 2020 we observed a sharp increase of $E_{abs}$ due to the observed sharp increase of OC to elemental carbon (EC) ratio. Our results show similar values of $E_{abs}$ to those found in the literature for similar background stations.

## 1  Introduction

The light-absorbing properties of atmospheric carbonaceous aerosols, i.e. black carbon (BC) and organic aerosols (OA), have been linked with a strong positive radiative forcing effect on Earth's energy budget (Liu et al., 2015; Zhang et al., 2018; Cappa et al., 2019). Recent scientific assessments(e.g. IPCC, 2021) on the global warming effect of anthropogenic agents have estimated that BC is the major aerosol contributing to the absorption of solar radiation from the ultraviolet to the infrared

part of the spectrum, with a direct radiative forcing (DRF) of $0.71 \pm 0.17$ $Wm^{-2}$ (Bond et al., 2013). However, the DRF of carbonaceous aerosols still presents large uncertainties given the limitations to constrain the spatial distribution, mixing state, and absorbing properties of these atmospheric aerosols in climate models (e.g. IPCC, 2021).

   BC particles can be mixed with less-absorbing and non-absorbing material through either external mixing, an heterogeneous mixture of internally homogeneous particles, or internal mixing, either an homogeneous mixture of internally homogeneous

particles or an heterogeneous mixture of particle composition and population (Bond and Bergstrom, 2006). The mixing state of BC with these aerosol particles determines its mass absorption cross-section (MAC), which is a spectral quantity relating the volumetric absorptive efficiency of a particle per unit mass, and is usually reported in square meters per gram $[m^2 g^{-1}]$. The MAC of pure BC (or elemental carbon, EC, depending on the measuring technique employed Lack et al., 2014) is rather well constrained. However, BC aggregates are rarely emitted as pure BC as they are usually co-emitted and internally mixed with

other source-dependent aerosols that can enhance the MAC of BC (e.g. Bond and Bergstrom, 2006; Knox et al., 2009; Lack and Cappa, 2010). Moreover, the absorption of radiation by less-absorbing particles externally mixed with BC, as absorbing OA also referred to as brown carbon (BrC) (Andreae and Gelencsér, 2006), also contributes to increase the measured absorption (Lack and Cappa, 2010). Different mixing states of BC particles were the cause for the regional differences found for the MAC in different background sites in Europe (Zanatta et al., 2016). The enhancement that this mixing produces in the resulting

observed MAC with respect to the theoretical pure BC MAC is defined as the absorption enhancement ($E_{abs}$). Understanding the relationship of $E_{abs}$ with the BC mixing state and the different aerosol species/sources is key to better parametrise the BC impact on radiative forcing (Jacobson, 2001; Bond et al., 2013). Whilst several studies assume $E_{abs}$ as only influenced by the internal mixing (e.g., Lack and Cappa, 2010), we used here an approach similar to Liu et al. (2015) where the spectral enhancement of light absorption by BC is considered as due to both the external and internal mixing of BC particles.

Externally mixed BrC particles also contribute to the enhancement of total absorption, although the absorption efficiency of BrC significantly decreases from UV moving into the visible (e.g. Moise et al., 2015; Laskin et al., 2015; Samset et al., 2018; Saleh et al., 2018; Saleh, 2020). BrC absorption coefficient values found in the literature display a large spatial variability (e.g.





Liu et al., 2015; Saleh et al., 2018; Saleh, 2020; Zhang et al., 2020) due to the specific organic aerosol sources and composition found for each site. These differences in OA composition result in different BrC MACs, since different OA from different

sources present absorption efficiencies variations (e.g. Saleh et al., 2018; Saleh, 2020; Zhang et al., 2020). Moreover, the MAC of different OA compounds shows different behaviour along the UV-visible range, hence the variation found in the influence of BrC on the absorption for this spectral range (Saleh et al., 2018; Saleh, 2020). The internal mixing contribution to $E_{abs}$ has been thoroughly studied both through core-shell models (Lack and Cappa, 2010) and by laboratory and field experiments (e.g. Cappa et al., 2019). The main differences in $E_{abs}$ values reported in literature were associated to different diameter of both BC

cores and shell in the case of model simulations, and to BC aging in the case of laboratory and field experiments. In fact, BC particles aging can be seen as a surrogate of the particles shell diameter since more ageing implies more coating layers (Lack and Cappa, 2010). Therefore, analyzing the influence on $E_{abs}$ of both internal and external BC mixing states is fundamental for a correct characterization of the aerosol particles light absorption and to better constrain modelling results (Liu et al., 2015).

Several laboratory studies, field measurements and modeling results can be found in the literature about $E_{abs}$ values (e.g.,

Lack and Cappa, 2010; Cappa et al., 2012; Liu et al., 2015; Zhang et al., 2018). However, the results fail to present an ubiquitous $E_{abs}$ value, with values ranging between almost no absorption enhancement ($E_{abs} \sim 1$, Cappa et al., 2012), to around a $50\%$ absorption increase as assumed by some climate models ($E_{abs} \sim 1.5$, Liu et al., 2015, , and references therein), up to values of more than a $100\%$, especially at the shorter wavelengths where the BrC externally mixed can largely contribute to the $E_{abs}$ ( e.g. Chen et al., 2017). As a consequence of the broad spectrum of values, several authors have suggested to treat $E_{abs}$ as a

regional specific parameter in climate models to account for the different sources or processes that may contribute to increase both the amount of BrC and the degree of BC internal mixing (Lack et al., 2012; Liu et al., 2015; Zhang et al., 2018).

To characterize the specific chemical species or sources that affect and to which extent the $E_{abs}$, simultaneous measurements of aerosol particle light-absorption at multiple wavelengths, elemental carbon (EC) concentrations, and particulate matter chemical composition analysis are needed. Although specific instrumentation (e.g. the single particle photometer, SP2; and

the soot particle aerosol mass spectrometer, SP-AMS) can be used for direct measurements of the chemical composition and internal mixing state of carbonaceous particles, their global implementation is sparse, thus impining a global characterization of $E_{abs}$ values. Using more simple yet robust monitors to obtain the source-dependent chemical composition influencing the absorption enhancement is possible (e.g. Zhang et al., 2018). The applied methodology consists on obtaining each measurement, e.g. chemical speciation, EC concentrations, light absorption, and aerosol aging through independent instruments, and

merging the results to the lowest timestamp possible.

Here we present an analysis of the BC light absorption enhancement measured at an urban station (Barcelona; BCN) and a regional station (Montseny; MSY) in the Western Mediterranean basin. The chemical analyses were performed using offline 24-hour filter measurements at both stations (between 2010 and 2020). In BCN online chemical composition measurements with a higher time resolution were also available (2018). In BCN, EC measurements were performed with a semi-continuous SUNSET

analyzer (Karanasiou et al., 2020) and submicron aerosol particles chemical composition measurements were performed with an Aerosol Chemical Speciation Monitor (ACSM; Via et al., 2021). Absorption measurements were performed with multiple-wavelength Aethalometer (AE33) and multi-angle absorption photometer (MAAP) instruments. In Sect. 3.1 we present an





overall analysis of both MAC and $E_{abs}$, showing the $E_{abs}$ seasonal variability and the contribution of both external and internal BC mixing states to $E_{abs}$. In Sect. 3.2, we performed an analysis of the relationship between the absorption enhancement and

the amount of non-refractory material available for coating with BC particles. Sect. 3.3 reports the results of a multi-linear regression analysis performed to identify the main sources/species responsible for the increase of $E_{abs}$ at both sites. Sect. 3.4 presents the influence of particle aging in the $E_{abs}$ values. Finally, we performed a trend analysis of $E_{abs}$ using the decade-long offline filter measurements available at both stations (Sect. 3.5). To the best of our knowledge, although some studies have shown the variability of MAC in the Mediterranean basin (e.g. Pandolfi et al., 2014b; Zanatta et al., 2016), this is the first study

of $E_{abs}$ in this region and one of the few studies of its kind performed in Europe (Liu et al., 2015; Zhang et al., 2018).

## 2    Methodology

### 2.1    Aerosol sampling sites and main characteristics

Measurements were performed at Barcelona – Palau Reial (BCN, urban background, Barcelona, $41°23'24.01''$N,$02°6'58.06''$E, 80 m a.s.l.), and Montseny (MSY, regional background, El Brull, $41°46'46''$N, $02°21'29''$E, 720 m a.s.l.) monitoring super-

sites (NE Spain). These measurement stations are characterized by aerosols with different physical and chemical properties. BCN urban station is located within the Barcelona metropolitan area of nearly 4.5 million inhabitants at a distance of about 5 km from the coast, and at 200 m distance from one of the most concured roads of the city (> 60k vehicles per day; City council of Barcelona). MSY regional station is located in a hilly and densely forested area within the Natural Park and Biosphere Reserve of Montseny, 50 km to the N–NE of the Barcelona and 25 km from the Mediterranean coast. A detailed charac-

terization of these measurement stations can be found in previous works (e.g. Querol et al. (2001); Rodrıguez et al. (2001); Reche et al. (2011); Brines et al. (2014, 2015); Ealo et al. (2018) for BCN; and Pérez et al. (2008); Pey et al. (2009); Pandolfi et al. (2011, 2014a, 2016) for MSY). These supersites are part of the Catalonian Air Quality Monitoring Network and are part of ACTRIS and GAW networks. Aerosol optical properties at BCN and MSY are measured following standard protocols (WMO/GAW, 2016).

Overall, the area of study is characterized by high concentrations of both primary and secondary aerosols from diverse emission sources (Rodríguez et al., 2002; Pandolfi et al., 2014a; Dayan et al., 2017; Rivas et al., 2020; Brean et al., 2020). Recently, Veld et al. (2021) presented the main aerosol sources in BCN and MSY by applying receptor modelling techniques to offline 24-hour speciated PM$_{2.5}$ samples collected during the period 2009-2018. The main sources identified from OA were the secondary OA (SOA), and from the secondary inorganic aerosols (SIA) they were sulphates, nitrates and ammonia.

Moreover, online ACSM measurements at BCN station (Via et al., 2021) have also shown that the organic aerosols are mainly dominated by secondary aerosols, also referred as oxygenated OA (OOA), as well as by hydrocarbon and cooking related OA (HOA and COA). Both Via et al. (2021) and Veld et al. (2021) have shown an increasing trend in the SOA as well as a reduction in the primary OA (POA) relative contribution to PM at BCN and MSY stations, mainly related with more restrictive pollutant emission policies and a larger amount of higher oxidative potential scenarios. The higher oxidative potential of the OA is

characterized by an increase in the relative proportion of the more-oxidized, MO-OOA, in comparison with the less-oxidized





OOA, LO-OOA (Via et al., 2021). Finally, a common characteristic of BCN and MSY measurement sites is that both are located in the proximity of North African deserts, thus both sites are heavily impacted by Sahara dust outbreaks (Querol et al., 2009; Yus-Díez et al., 2020). For this reason,in order to avoid the interference due to dust absorption, we filtered out scenarios when the sites were under the influence of dust outbreaks (following the European Commission guidelines; European Commission, 130    2011).

### 2.2    Absorption coefficients and EC measurements

At both measurement sites, aerosol particle absorption coefficients ($b_{abs}$) at 637 nm were obtained with multi angle absorption photometers (MAAP, Model 5012, Thermo Inc., USA, Petzold and Schönlinner, 2004). Moreover, in BCN absorption measurements were also performed with a multi-wavelength aethalometer (model AE33, Magee Scientific, Aerosol d.o.o. Drinovec 135    et al., 2015) at seven different wavelengths (370, 470, 520, 590, 660, 880 and 950 nm). The MAAP absorption coefficients at 637 nm (Müller et al., 2011) were derived by the internal MAAP software using a radiative transfer model from the measurements of transmission of light through the filter tape and backscattering of light at two different angles. MAAP measurements were obtained with a 1 min time resolution at a flow rate of 5 l/min and with a $PM_{10}$ inlet cut-off. The AE33 $b_{abs}$ coefficients in BCN were derived with the same time resolution and flow rate as the MAAP and with $PM_{2.5}$ inlet. The aethalometer filter 140    loading effect was corrected online by the dual-spot manufacturer correction (Drinovec et al., 2015), and the multiple scattering correction constant, C, was set to 2.44, as obtained by Yus-Díez et al. (2021). Absorption measurements errors of 12% and 15% were set for the MAAP and AE33, respectively (Petzold and Schönlinner, 2004; Rigler et al., 2020).

Semi-continuous EC measurements were obtained in BCN by means of a Semi-Continuous OC:EC aerosol analyzer (Sunset Laboratory Inc.) with a $PM_{2.5}$ inlet cut-off at a flow rate of 8.0 l/min, a measuring interval of 3 hours using the EUSAAR2 145    protocol, with a measurement error of 10% (Karanasiou et al., 2020). The device was equipped with a C parallel-plate diffusion denuder to remove VOCs that can be adsorbed on quartz fibre filters and cause positive artefacts in the OC measurement (Viana et al., 2006).

$PM_{10}$ 24-hour offline filter samples were collected at both BCN and MSY on 150 mm quartz micro-fibre filters (Pallflex 2500 QAT-UP) using high-volume samplers (DIGITEL DH80 at 30 m$^3$ h$^{-1}$). The 24-hour average concentrations of major 150    and trace element, and soluble ions (determined following the procedure by Querol et al. (2001)), as well as those of organic (OC) and elemental (EC) carbon (by a thermal-optical carbon analyser, SUNSET, following the EUSAAR2 protocol (Cavalli et al., 2010)) were obtained from these offline filter samples and were estimated to have a measurement error of 10%.

### 2.3    Submicron non-refractory PM chemical composition measurements and OA source apportionment

A Quadrupole Aerosol Chemical Speciation Monitor (Q-ACSM, Aerodyne Research Inc.) was deployed in BCN for chemical 155    speciation of submicrometric particles at a flow rate of 3 l/min. The incoming particles go through an aerodynamical lens transmitting particles of aerodynamic diameters from 75 to 650nm. Then, these particles are vaporized, ionized by hard-electron impact and fragmentated and the resulting fragments are analyzed by a quadrupole mass spectrometer. The instrument can provide using a fragmentation table (Allan et al., 2004) the concentrations of non-refractory $PM_1$ species (OA, sulphate, nitrate,





ammonia and chloride) with 30 min resolution and a 12-120 Th OA spectra matrix. The software used for data acquisition
and treatment was provided by Aerodyne Inc. (versions 1.6.0.0 and 1.6.1.1, respectively) and implemented in the Igor Pro
(Wavemetrics, Inc.) environment.

The OA matrices retrieved were used as input for Positive Matrix Factorization analysis (PMF; Paatero and Tapper, 1994),
applied using multi-linear engine (ME-2) (Paatero, 1999) to differentiate the different OA sources. A detailed description of
the OA sources detected in BCN and used in this work can be found in Via et al. (2021). Briefly, the OA sources in BCN
were: Cooking-like OA (COA), Hydrocarbon-like OA (HOA), Biomass Burning OA (BBOA), Less-Oxidized Oxigenated OA
(LO-OOA) and More Oxidized Oxygenated OA (MO-OOA).

### 2.4  Determination of the absorption enhancement, $E_{abs}$

Here, similarly to Zhang et al. (2018), we derived $E_{abs}$ as the ratio between the measured ambient mass absorption cross-
section (MAC) calculated at the different wavelengths available from the AE33 and the MAAP, and the reference MAC value
of pure BC.

The observed ambient MAC defines the contribution to the measured absorption coefficients from BC particles internally
and externally mixed with organic and inorganic species that can contribute positively to the measured absorption (Bond
et al., 2013). The ambient MAC measurements were obtained as the ratio of the light absorption coefficients ($b_{abs}$) at a given
wavelength, $\lambda$, and the elemental carbon (EC) concentrations obtained with the Sunset analyzer, either online or offline,

$$MAC^\lambda = \frac{b_{abs}^\lambda}{[EC]}. \tag{1}$$

The enhancement of the absorption due to both internal and external mixing of the BC particles can be quantified by
normalizing the measured ambient $MAC^\lambda$ with a reference value for pure BC, $MAC_{ref}^\lambda$. As already stated, we have applied
here the same methodology to determine $E_{abs}$ as in Zhang et al. (2018) and Liu et al. (2015) by calculating the ambient MAC
through equation (1) applied to the 7 AE33 wavelengths. Thus, since BrC absorbs more efficiently at the shortest wavelengths
(370-470 nm mostly) but not at 880 nm, the observed $E_{abs}$ at the shortest wavelengths includes the lensing-driven enhancement
and the enhancement induced by semi-volatile BrC (e.g. Liu et al., 2015), whereas the observed $E_{abs}$ at 880 nm represents the
lensing-driven enhancement only.

$$E_{abs}^\lambda = \frac{MAC^\lambda}{MAC_{ref}^\lambda}. \tag{2}$$

The reference MAC, $MAC_{ref}^\lambda$, can be obtained either from the literature (Bond and Bergstrom, 2006), or from the experi-
mental data. There are two experimental approaches for obtaining $MAC_{ref}^\lambda$: 1) by using denuded measurements that evaporate
the semi-volatile organic and inorganic species thus allowing for a measurement of the pure BC absorption (e.g. Liu et al.,
2015), and 2) by using as $MAC_{ref}^\lambda$ the intercept of the relationship between the ambient $MAC^\lambda$ and the OC:EC ratio. In this





letter case $\text{MAC}^\lambda_{ref}$ is the MAC value obtained when the OC:EC ratio is equal to 0 (Zhang et al., 2018). Here we have preferentially used the second method (intercept) to determine the reference value for $\text{MAC}^\lambda_{ref}$ (Fig. S1-S3). For this purpose we

used a Deming regression fit taking into account the propagation of errors from the absorption and OC:EC measurement errors. Additionally, we also used the literature MAC reference value to calculate $\text{E}_{abs}$, i.e. $7.5 \pm 1.2$ ($\text{m}^2\text{g}^{-1}$) at 550 nm (Bond and Bergstrom, 2006), which was extrapolated to each AE33 wavelength assuming an Absorption Ångström Exponent (AAE) of 1.

Online $\text{MAC}^\lambda$ and $\text{E}^\lambda_{abs}$ values at BCN were obtained using AE33 $\text{b}^\lambda_{abs}$ coefficients averaged to the semi-continuous Sunset

OC:EC measurements time stamp (3h) during 2018 when the Q-ACSM measurements were also available. Offline MAC values at both BCN and MSY were obtained by using 24-hour average $\text{b}_{abs}$ coefficients from the MAAP at 637 nm, and 24h OC:EC concentrations from $\text{PM}_{10}$ filters (2010-2020). Figure S4 shows the obtained $\text{MAC}^\lambda_{ref}$ values for both BCN and MSY stations for all the wavelengths available from both online (BCN) and offline (BCN and MSY) measurements. Data in Figure S4 were grouped into two periods: a cold period from December to May, and a warm period from June to October. As

shown later, measurements were grouped into these two distinct periods due to the source apportionment results from Q-ACSM measurements in BCN, since the BBOA-like compounds were only detected during winter and spring (Via et al., 2021). Offline $\text{MAC}^\lambda_{ref}$ values at both stations showed a good agreement with the reference theoretical value obtained in Bond and Bergstrom (2006), whereas the online measurements obtained with the AE33 were higher through-out the whole spectrum. Similar higher than the theoretical $\text{MAC}^\lambda_{ref}$ values have also been reported in other studies, such as in Zhang et al. (2018).

Furthermore, we have assumed here that BrC particles do not absorb at 880 nm (Kirchstetter et al., 2004) and that the measured absorption at this wavelength was only driven by the BC internally mixed particles (i.e. the lensing effect). Moreover, we assumed that the lensing-driven absorption enhancement for BC particles was wavelength independent (Lack and Cappa, 2010; Liu et al., 2015; Zhang et al., 2018). Thus, the $\text{E}_{abs}$ at 880 nm can be described as follows (Eq. 3):

$$E^{880}_{abs} = E^{880}_{abs,int} = E_{abs,int}, \tag{3}$$

where the subscript $\text{E}_{abs,int}$ only refers to the absorption enhancement due to BC internal mixing.

Therefore, the absorption enhancement due to externally mixed particles at a given wavelength, $\text{E}^\lambda_{abs,ext}$, was obtained as the difference between the measured total ambient absorption enhancement and the absorption enhancement due to the internal mixing,

$$E^\lambda_{abs,ext} = E^\lambda_{abs} - E_{abs,int}. \tag{4}$$

## 2.5 Chemical fractions contribution to $\text{E}_{abs}$

Submicron chemical composition from ACSM in BCN and offline 24h chemical speciated data from filter analyses at both BCN and MSY were used to determine the influence of the material available for BC coating on $\text{E}_{abs}$. Chemical speciated data





were used to calculate the total amount of non-refractory particulate matter (NR-PM$_{10}$ for offline measurements and NR-PM$_1$ for online measurements) mass concentration. The NR-PM to EC concentration ratios were then calculated as follows:


$$R_{NR-PM} = \frac{\sum_i [NR - PM]_i}{[EC]};$$ (5)

Thus, the calculated R$_{\mathrm{NR-PM}}$ represents a proxy for the amount of non-refractory material available for mixing with the BC particles (Cappa et al., 2019). The chemical species and groups of compounds taken into account using the online ACSM continuous measurements were: $[HOA]$, $[COA]$, $[LO-OOA]$, $[MO-OOA]$, $[SO_4^{2-}]$, $[NO_3^-]$, $[NH_4^+]$ and $[Cl^-]$, plus $[BBOA]$ during the cold period; whereas from the 24-hour filters they were: $[OA]$, $[SO_4^{2-}]$, $[NO_3^-]$, $[NH_4^+]$, $[Cl^-]$.

Moreover, online submicron chemical composition data and OA source apportionment from ACSM in BCN were used to determine the species that mostly contributed to E$_{\mathrm{abs}}$. For this, a multivariate linear regression (mlr) analysis was employed to solve the following equation:

$$E_{abs} = E_0 + m_1[q_1] + m_2[q_2] + ... + m_z[q_z];$$ (6)

where E$_0$ is the intercept, $m_i$ (where $i = 1, ..., z$) are the regression coefficients, i.e. the relative contribution of each chemical
fraction to E$_{\mathrm{abs}}$, and $[q_i]$ are the dependent variables of the mlr, i.e. the ratios of each chemical fraction/source normalized to the EC concentration. Note that E$_0$ should be equal to 1 (i.e. no absorption enhancement) when all the ratios are equal to 0 in eq. (6). In order to perform a more robust mlr analysis and to reduce the effect of outliers, data points lower than the 5th and higher than the 95th percentiles were excluded from the analysis.

## 3   Results

### 3.1   Site specific MAC and E$_{\mathbf{abs}}$ analysis

The median values of the ambient BC MAC and E$_{\mathrm{abs}}$ at different wavelengths from both online and offline measurements at BCN and MSY are reported in Table 2 and Table 1, respectively. For the long-term offline measurements, the MAC at 637 nm was $9.67 \pm 2.55$ m$^2$g$^{-1}$ at BCN urban background station, and $13.10 \pm 4.47$ m$^2$g$^{-1}$ at MSY regional background station. The BC MAC at MSY showed a higher median value compared to BCN (cf. Table 1) due to the fact that BC particles reaching
the regional station had more time to gather material for coating. Moreover, the frequency distribution of the MAC values at MSY was less left-skewed and more right-skewed compared to BCN (see Fig. S5). For the intensive online measurements in BCN, the MAC ranges between $7.55 \pm 2.06$ at 950 nm and $22.74 \pm 6.98$ at 370 nm (Table 1). The values at 660 nm for the online measurements and 637 nm for the offline measurements were around $10.85 \pm 2.98$, and $9.67 \pm 2.55$ in BCN, and $13.10 \pm 4.47$ in MSY, which were within the range of MAC values reported by Zanatta et al. (2016) for similar station backgrounds.
The observed increase of MAC with decreasing wavelength was expected due to both the increase of the energy radiation and the larger influence of the externally mixed BrC particles at shorter wavelengths. In fact, the effect on MAC of externally





mixed BrC particles in BCN is visible in Fig. S5 where the frequency distribution of MAC at 370 nm showed a much more pronounced tile toward higher values compared to the MAC at 880 nm.

**Table 1.** Observed MAC ($m^2 g^{-1}$) values obtained using online techniques via AE33 and Sunset online EC measurements at BCN (BCN$_\text{on}$), and offline at BCN and MSY via MAAP and offline EC measurements on 24-hour filters (X$_\text{off}$).

|  | $\lambda$ (nm) | MAC |
|---|---|---|
| **Online BCN** | 370 | $22.74 \pm 6.98$ |
|  | 470 | $17.23 \pm 4.80$ |
|  | 520 | $14.84 \pm 4.16$ |
|  | 590 | $12.63 \pm 3.50$ |
|  | 660 | $10.85 \pm 3.02$ |
|  | 880 | $7.92 \pm 2.16$ |
|  | 950 | $7.55 \pm 2.06$ |
| **Offline BCN** | 637 | $9.67 \pm 2.55$ |
| **Offline MSY** | 637 | $13.10 \pm 4.47$ |

The averaged multi-wavelength absorption enhancement values from both online and offline measurements at BCN and
MSY are shown in Table 2. The online measurements at 880 nm in BCN led to a median value of E$_\text{abs}$ of $1.28 \pm 0.36$, whereas it increased to $1.45 \pm 0.51$ at 370 nm. For the offline measurements, the median E$_\text{abs}$ values at 637 nm were $1.42 \pm 0.40$ and $2.00 \pm 0.75$ at BCN and MSY, respectively. As reported in Table 2, the E$_\text{abs}$ values from online and offline measurements in BCN were rather similar ($1.31 \pm 0.38$ at 660 nm online and $1.42 \pm 0.40$ at 637 nm offline), and the observed difference was likely due to both the different periods and the different instrumentation used for the calculation of E$_\text{abs}$. As already observed
for the MAC, the higher E$_\text{abs}$ at the regional MSY station was due to ageing of BC particles during the transport toward the regional station. The E$_\text{abs}$ reported in Table 2 were calculated from eq. (2) using as MAC$_{ref}^{\lambda}$ the intercept values from the Deming regression fits reported in Fig. S1-S4 for online AE33 and offline BCN and MSY, respectively. If the theoretical reference MAC (Bond and Bergstrom, 2006) of uncoated BC was used, the overall median E$_\text{abs}$ values were higher for the online measurements at BCN ($1.59 \pm 0.51$ at 880 nm and $1.91 \pm 0.62$ for 370 nm), although within the uncertainty, and rather
similar for the offline measurements at BCN and MSY, with E$_\text{abs}$ values at 637 of $1.43 \pm 0.44$ and $1.92 \pm 0.76$, respectively (Table 2). The values found at the urban BCN area using the experimental (theoretical) reference MAC at 880 nm were similar (higher) to those observed in the literature for the same wavelength at rural/suburban areas (Liu et al., 2015; Zhang et al., 2018). The mean E$_\text{abs}$ value observed at the regional background MSY station was also similar to the values reported in literature for rural areas (e.g., Cui et al., 2016).
Figure 1 shows the density distribution of E$_\text{abs}$ for the cold (from December to May) and warm (from June to October) seasons from both online measurements at BCN and offline measurements at BCN and MSY (Fig. 1). The median values of



**Table 2.** Overall, and cold and warm period average $E_{abs}$ values for both multi-wavelength online measurements at BCN, and offline measurements at the near-infrared at BCN and MSY station.

| | $\lambda$ (nm) | $\mathbf{E_{abs,exp}}$ Overall | cold | Warm | $\mathbf{E_{abs,theory}}$ Overall | cold | Warm |
|---|---|---|---|---|---|---|---|
| **Online BCN** | 370 | 1.45 ± 0.51 | 1.67 ± 0.57 | 1.31 ± 0.35 | 1.91 ± 0.69 | 2.11 ± 0.79 | 1.79 ± 0.53 |
| | 470 | 1.38 ± 0.43 | 1.53 ± 0.50 | 1.27 ± 0.29 | 1.85 ± 0.61 | 2.00 ± 0.71 | 1.76 ± 0.45 |
| | 520 | 1.35 ± 0.41 | 1.47 ± 0.48 | 1.27 ± 0.30 | 1.76 ± 0.58 | 1.91 ± 0.67 | 1.67 ± 0.43 |
| | 590 | 1.33 ± 0.39 | 1.42 ± 0.46 | 1.27 ± 0.29 | 1.70 ± 0.55 | 1.83 ± 0.65 | 1.61 ± 0.40 |
| | 660 | 1.31 ± 0.38 | 1.39 ± 0.45 | 1.26 ± 0.29 | 1.63 ± 0.54 | 1.76 ± 0.63 | 1.55 ± 0.39 |
| | 880 | 1.28 ± 0.36 | 1.33 ± 0.43 | 1.25 ± 0.28 | 1.59 ± 0.51 | 1.69 ± 0.60 | 1.52 ± 0.37 |
| | 950 | 1.28 ± 0.36 | 1.33 ± 0.43 | 1.25 ± 0.28 | 1.59 ± 0.51 | 1.69 ± 0.60 | 1.52 ± 0.37 |
| **Offline BCN** | 637 | 1.42 ± 0.40 | 1.41 ± 0.39 | 1.45 ± 0.40 | 1.43 ± 0.44 | 1.42 ± 0.43 | 1.43 ± 0.44 |
| **Offline MSY** | 637 | 2.00 ± 0.75 | 1.82 ± 0.63 | 2.24 ± 0.79 | 1.92 ± 0.76 | 1.73 ± 0.66 | 2.02 ± 0.81 |

the season-dependent frequency distributions of $E_{abs}$ were reported in Table 2. The long-term offline measurements led, on average, to similar $E_{abs}$ values at 637 nm in BCN during the warm (1.45 ± 0.40) and cold (1.41 ± 0.39) seasons. However, at MSY $E_{abs}$ values were larger during the warm period (2.24 ± 0.79) compared to the cold period (1.82 ± 0.63), mainly due to
the increase in the secondary organic aerosol formation (Fig. S6 and S7) which was mostly driven by the increase in biogenic volatile organic compounds (VOCs) during the warm season (Seco et al., 2013; Veld et al., 2021). Thus, as shown later in more detail, OA, and especially SOA, contributed strongly to the BC lensing-driven absorption enhancement, especially at the regional station.

     The online measurements performed with AE33 aethalometer allowed for a multi-wavelength analysis of $E_{abs}^{\lambda}$. Figure 1a
shows a seasonal decoupling of $E_{abs}$ between the near-ultraviolet and the infrared wavelengths: whilst in the warm period $E_{abs}$ remained similar for all the wavelengths, during the cold period there was an increase of $E_{abs}$ towards the shorter wavelengths. This different amplification of the absorption enhancement at the near-ultraviolet can be associated with a larger presence of BrC-like compounds (e.g. BBOA from winter biomass burning) (Fig. S6 and S7) during the cold period (Via et al., 2021), which present larger mass absorption cross-sections and contribution to absorption at these wavelength range (e.g. Lack et al.,
2012; Qin et al., 2018; Saleh et al., 2018; Saleh, 2020; Kasthuriarachchi et al., 2020; Zhang et al., 2020). Figure S8 shows that by using the theoretical reference MAC the density distribution of $E_{abs}$ was similar, although the differences between the cold and warm period were not that great. This was attributed to the fact that using a theoretical MAC does not take into account the different seasonal-dependent contributions of OA sources.





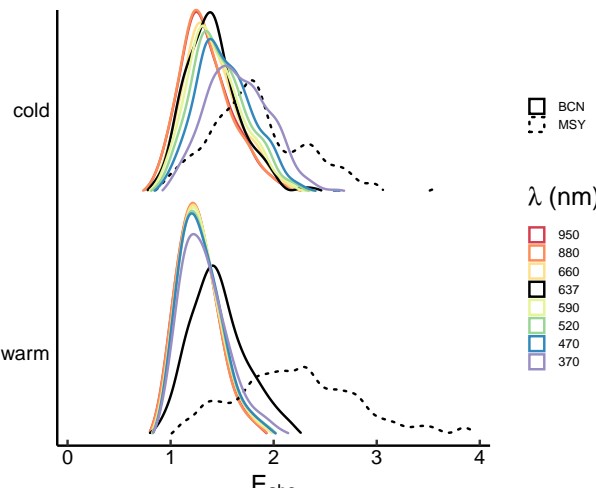

**Figure 1.** Seasonal frequency distributions of $E_{abs}$ at BCN for the multiple wavelengths measured with the AE33 (colored solid lines for 370, 470, 520, 590, 660, 880 and 950 nm), and at BCN (black solid line) and MSY (black dash line) measured with a MAAP 637 nm and offline filters.

### 3.1.1 $E_{abs}$ dependence on the mixing state

As already stated, ambient BC particles can be either externally or internally mixed with other aerosols (Bond and Bergstrom, 2006). In order to separate the relative contributions to $E_{abs}$ of these two mixing states, i.e. external ($E_{abs,ext}$) and internal ($E_{abs,int}$), we used the multi-wavelength AE33 and the semi-continuous OC:EC measurements obtained in BCN (see Sect. 2.4). We assumed that the $E_{abs}$ at the near-infrared (880 nm) was only produced by the internal mixing of BC particles, whereas at the short-UV (370 nm) the $E_{abs}$ is due to both the internal and external mixing of BC particles. Given the spectral characteristic of BrC absorption, the contribution to $E_{abs}$ due to external mixing was the highest at 370 nm compared to the other AE33 wavelengths.

The contribution due to the internal mixing ($E_{abs,int}$) had a constant value for all the wavelengths of $1.28 \pm 0.36$. Overall, this $E_{abs,int}$ value represented between 100% and 88% of the total $E_{abs}$ at 880 nm and 370 nm, respectively. Thus, the BrC externally mixed particles represented a non-negligible fraction of the total $E_{abs}$ at near-ultraviolet wavelengths (Table S1), increasing from $0.03 \pm 0.04$ (2.2%) at 660 nm up to $0.17 \pm 0.18$ at 370 nm (11.7%). If we also consider the contribution of pure BC (without mixing) to the measured total absorption, then the BrC externally mixed represented between 2.37% and 15.2% of the total absorption at 660 nm and 370 nm, respectively, whereas the internal mixing contribution ranged between 19.10% and 16.59% at 660 nm and 370 nm, respectively. The remaining absorption was due to pure BC particles. If the theoretical $MAC_{ref}$ was used for the calculation of the absorption enhancement in BCN, then the absorption enhancement due to the BC internal and external mixing at 370 nm increased (as it did the overall average absorption enhancement) to $1.59 \pm 0.51$ and $0.32 \pm 0.23$, respectively, and therefore the external mixing represented around 16% of the total $E_{abs,370}$.





Although BCN is an urban background station with a non-predominant contribution from biomass burning, the contribution
to absorption from other potential BrC sources cannot be excluded. Since biomass burning emissions are higher during the cold
season, we have found that during this season compared to the warm period there was a small increase of the total absorption
enhancement due to the internal mixing (6.4%), and a more significant increase of contribution of the external mixing $E_{abs}$,
which increased from 0.06 during the warm period to 0.34 during the cold season (Table S1), i.e. from representing a 4.6% to
20.3% of the total $E_{abs}$, respectively. In fact, biomass burning is not the only source contributing to the presence of BrC in the
atmosphere during the colder months (e.g. Zhang et al., 2020), in fact, as shown later, other OA sources also contribute to $E_{abs}$.

## 3.2    $E_{abs}$ dependence on $R_{NR-PM}$ content

Here we analyzed the relationship between the $E_{abs}$ and the amount of material available for mixing with BC particles
($R_{NR-PM}$, c.f. Sect. 2.5). As commented in Sect. 3.1, the variability of $E_{abs}$ with the seasons can be attributed to the dif-
ferences in the OA composition and concentration levels. Cappa et al. (2019) have shown the discrepancies between model,
laboratory and field studies in the behaviour of $E_{abs}$ with $R_{NR-PM}$. Indeed, Figure 1 in Cappa et al. (2019) shows that some
studies reported only a slight increase of $E_{abs}$ with the amount of the coating material (Cappa et al., 2012), whereas others
measured a larger increase of $E_{abs}$ for high concentrations of $R_{NR-PM}$ (Liu et al., 2015; Peng et al., 2016). The authors argued
that the differences could be associated to the ageing state, volatility, and amount of coating of the particles, as well as to the
apportionment of external mixing to $E_{abs}$.

Figure 2 shows $E_{abs}$ values as a function of $R_{NR-PM}$ at BCN (online and offline) and MSY (offline). Overall, we observed
an exponential increase of the absorption enhancement with the amount of NR-PM coating material available for mixing (Fig.
2), which was consistent with some of the observed behaviour found in the literature (e.g. Fig. 1 in Cappa et al., 2019).

As shown in Fig. 2, the $R_{NR-PM}$ binned values from offline measurements at MSY spanned from around 15 up to around
55 µg/m$^{-3}$, whereas in BCN $R_{NR-PM}$ values were between around 3.5 and 20 µg/m$^{-3}$. As a consequence, the $E_{abs}$ at MSY
reached values up to around 3.25, whereas in BCN $E_{abs}$ values remained lower than 2. Thus, the higher $R_{NR-PM}$ at MSY
implied that more material was available for BC coating at the regional site compared to BCN, thus leading to higher $E_{abs}$ at
MSY. Moreover, the lowest $E_{abs}$ at MSY from binned data in Fig. 2 was around 1.3 indicating that on average BC particles
reaching MSY station have undergone a longer aging processes and were already coated, whereas in BCN freshly emitted
still-not-mixed BC particles were frequently measured, as denoted by $E_{abs}$ values closer to 1. For the online measurements in
BCN, $R_{NR-PM}$ showed a larger range of values (from around 4 to 40 µg/m$^{-3}$) compared to the offline measurements because
of the higher time resolution of on-line measurements allowing measuring events characterized by lower or higher $R_{NR-PM}$
compared to the 24-hour offline measurements. The higher $E_{abs}$ in BCN at 637 nm compared to $E_{abs}$ at 370 nm was mostly
associated to the different inlets and, to a lesser extent, to the different periods used for the online and offline measurements.
In fact, as shown in Fig. S9, the mean $E_{abs}$ calculated from offline measurements in BCN using the same period as for the
online measurements (2018) was closer to the $E_{abs}$ values obtained from online measurements at 880 nm and 370 nm. As
shown in Fig. 2, the $E_{abs}$ values from the online and offline datasets in BCN, showed similar trends for smaller concentrations
of $R_{NR-PM}$. However, as the amount of mixing material increased, the offline method increased at a higher rate, reaching



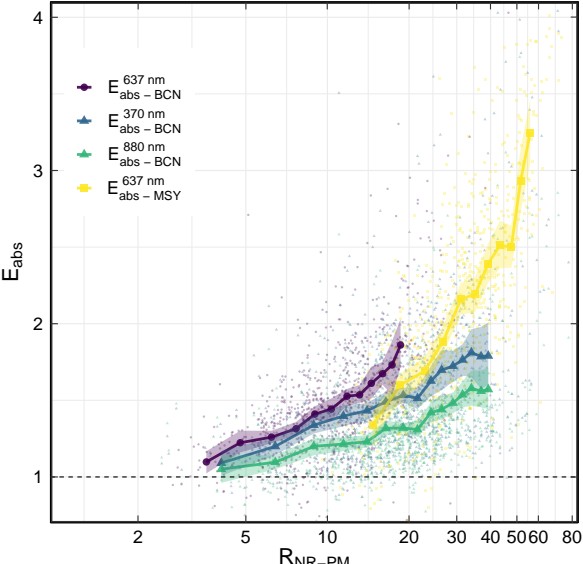

**Figure 2.** Absorption enhancement, $E_{abs}$, as a function of the non-refractory PM to EC ratio at both Barcelona (BCN) and Montseny (MSY) station. The offline $E_{abs}$ measurements with a $PM_{10}$ inlet at 637 nm were used for both BCN and MSY. Also, at BCN, online $E_{abs}$ measurements with a $PM_{2.5}$ inlet at the short-UV (370 nm) and near-IR (880 nm) wavelengths were used. The scatter points represent all the measurements, whereas the marked points show the mean of each bin, whilst the shadow of the line represent the standard deviation of each bin.

higher values for the largest $R_{NR-PM}$ measurements. This behaviour was also observed when only the 2018 year period was used for the calculation of $E_{abs}$ from offline measurements (cf. Fig. S9). These different trends between online and offline $E_{abs}$ versus $R_{NR-PM}$ were probably due to two main factors: first, the offline measurements were made with a $PM_{10}$ inlet vs the $PM_{2.5}$ inlet of the online method, hence coarse nitrates and other coarse particles could have influenced $E_{abs}$, and, second, the

large annual variability observed for the offline $E_{abs}$ measurements (see Fig. S9) could have also contributed to the observed difference.

The availability of multiple wavelength absorption enhancement values at BCN station allowed a multi-wavelength analysis of $E_{abs}$ versus $R_{NR-PM}$. Since $E_{abs}$ at 880 nm is influenced solely by the internal mixing of BC particles, the comparison of $E_{abs}$ at 880 nm and 370 nm, when the external mixing influence on the absorption is the highest, showcased the influence

that the mixing state had on the relationship between $E_{abs}$ and $R_{NR-PM}$. Figure 2 shows similar values for both wavelengths for the lowest amount of coating material, but as $R_{NR-PM}$ increased the absorption enhancement at 370 nm increased more rapidly than at 880 nm, mostly due to the fact that as $R_{NR-PM}$ increases the contribution of BrC externally mixed becomes larger.

If a theoretical MAC reference was used instead, the $E_{abs}$ values as a function of $R_{NR-PM}$ at BCN (online and offline) and

MSY (offline) showed the same relationship (Fig. S10), albeit larger values were observed, specially for the AE33 measure-





**Table 3.** Multivariate linear regression analysis coefficients and standard deviation of the chemical fraction influence on the absorption enhancement, $E_{abs}$, for Barcelona at both 370 and 880 nm wavelengths using the Q-ACSM chemical sources (Via et al., 2021)

|  | 370 nm | | 880 nm | |
| --- | --- | --- | --- | --- |
|  | **Cold** | **Warm** | **Cold** | **Warm** |
| **Intercept** | $1.097 \pm 0.062$ | $1.112 \pm 0.028$ | $1.003 \pm 0.048$ | $1.109 \pm 0.022$ |
| **HOA:EC** | $0.195 \pm 0.099$ | $0.092 \pm 0.038$ | $0.126 \pm 0.077$ | $0.019 \pm 0.029$ |
| **BBOA:EC** | $0.175 \pm 0.058$ |  | $-0.062 \pm 0.044$ |  |
| **MO-OOA:EC** | $0.044 \pm 0.021$ | $0.010 \pm 0.007$ | $0.040 \pm 0.016$ | $-0.009 \pm 0.005$ |
| **LO-OOA:EC** | $0.161 \pm 0.064$ | $-0.001 \pm 0.006$ | $-0.012 \pm 0.049$ | $0.006 \pm 0.005$ |
| **Sulphate:EC** | $-0.003 \pm 0.017$ | $0.010 \pm 0.004$ | $0.012 \pm 0.013$ | $0.017 \pm 0.003$ |
| **Nitrate:EC** | $-0.011 \pm 0.010$ | $0.087 \pm 0.011$ | $-0.006 \pm 0.008$ | $0.060 \pm 0.009$ |
| **COA:EC** | $0.106 \pm 0.040$ | $0.032 \pm 0.022$ | $0.035 \pm 0.030$ | $0.044 \pm 0.018$ |

ments. These larger values were mostly due to the fact that the experimentally used MAC was higher than the theoretical ones (Fig. S4), resulting in larger $E_{abs}$ values (Fig. S8).

### 3.3 Aerosol sources contribution to $E_{abs}$

The material available for coating on BC particles, which determines its absorption enhancement properties (see Fig. 2), is formed by an array of different chemical compounds from different sources as a result of a succession of physical and chemical processes in the atmosphere. These different chemical compounds can exert different responses on $E_{abs}$ (e.g. Zhang et al., 2018) and can increase the BC $E_{abs}$ depending on their relative amount compared to BC as shown in Fig. 2 and the literature (Zhang et al., 2018; Cappa et al., 2019, and therein).

Here we analyzed the contribution of different OA sources and chemical species, as sulphate and nitrate, to the $E_{abs}$ calculated in Barcelona from online measurements via a multi-variate linear regression analysis. The OA sources were obtained by means of a PMF analysis on the Q-ACSM data in BCN and were published by Via et al. (2021). Given the differences in the seasonality that the OA sources can present, as also observed in BCN (cf. Fig. S7; Minguillón et al., 2015; Via et al., 2021), and given the observed $E_{abs}$ seasonality (Fig. 2), we applied the MLR analysis separately to the warm and cold periods. Furthermore, in order to separate the contribution of the different BC mixing states, we performed the mlr analyses at the same wavelengths as in Sect. 3.2, namely 370 and 880 nm.

The results of the mlr analysis were reported in Table 3. Table 3 shows that overall, regardless of the season and the wavelength considered, the main contributors to $E_{abs}$ in BCN were Hydrocarbon-like OA (HOA), associated to the emissions from traffic, and cooking-related OA (COA). These two sources contributed 12 and 14 %, respectively, to the measured OA mass concentration (Via et al., 2021). Thus, in BCN, HOA and COA increased $E_{abs}$ both by contributing to the BC coating (880 nm)





and by acting as BrC species externally mixed with BC (370 nm), as suggested by the higher coefficients observed for these two OA sources at 370 nm compared to 880 nm (cf. Table 3). A major source of BC in BCN was traffic (Pandolfi et al., 2016; Via et al., 2021), thus likely explaining the high potential of HOA particles to contribute to to $E_{abs}$. Moreover, some recent studies have also shown that HOA particles in urban environments can potentially have a high absorption efficiency in the UV-VIS (Qin et al., 2018; Chen et al., 2020; Kasthuriarachchi et al., 2020). Regarding the COA particles, some studies have

shown that this OA source has a lower absorption efficiency compared to HOA particles (e.g. Qin et al., 2018; Kasthuriarachchi et al., 2020; Chen et al., 2020). This was in agreement with the lower coefficients observed for for COA compared to HOA (Table 3). In addition, during the cold period both BBOA and LO-OOA presented large positive contributions to $E_{abs}$ at 370 nm, whereas at 880 nm the contribution to $E_{abs}$ from these two sources was negative. For the warm period the contribution of LO-OOA was also very low at 880 nm and negative at 370 nm whereas, as already stated BBOA did not contribute in summer.

Zhang et al. (2018) also found a negative contribution to $E_{abs}$ for BBOA at 880 nm. The higher coefficients observed for these sources (HOA, COA, BBOA and LO-OOA) at 370 nm compared to 880 nm highlighted the potential of these OA sources to act as BrC species externally mixed with BC. MO-OOA particles also contributed positively to $E_{abs}$ at both 370 nm and 880 nm especially during the cold season likely due to the higher relative contribution of MO-OOA to OA observed in this season compared to the warm period (Via et al., 2021). Recently, Kasthuriarachchi et al. (2020) reported higher absorption efficiency

in the UV-VIS range for LO-OOA particles compared to MO-OOA particles likely due to the photo-degradation chemistry (photo-bleaching) of BrC chromophores in this aged MO-OOA fraction. The inorganic aerosol components presented a higher variability with regards to their contribution to $E_{abs}$, with sulphates, and especially nitrates, becoming an important source of coating during the warm period, whilst presenting a low impact during the colder period.

Figure 3 shows the contribution to $E_{abs}$ from the OA sources and inorganic species included in the mlr analysis as $R_{NR-PM}$

increases. The contributions to $E_{abs}$ reported in Figure 3 were calculated as the product between the OA sources and inorganic species mass concentrations (provided by Via et al., 2021) to EC ratios and the coefficients reported in Table 3. As shown in Fig. 3, during the cold season in BCN the absolute contribution of OA to $E_{abs}$ was much higher compared to the contribution from inorganic aerosols (nitrates and sulphates here) at both 370 nm and 880 nm (Figs. 3a,c). HOA and MO-OOA were the major sources contributing to the lensing-driven absolute BC absorption enhancement at 880 nm in BCN during the cold season.

Conversely, the absolute contribution of LO-OOA and COA to $E_{abs}$ was the highest at 370 nm suggesting their importance as BrC source in the area under study. During the warm period, as already noted, $E_{abs}$ was lower compared to the cold period. The major difference compared with the cold period was that the contribution of secondary inorganic aerosols increased. It was notable the contribution at 880 nm of sulphates as BC lensing-driving species. Furthermore, Fig. 3 also shows that when the contributions were negative, these did not have a large effect upon the overall $E_{abs}$.

## 3.4 Atmospheric aging influence on $E_{abs}$

Atmospheric aging and oxidation of OA particles have been shown to have an important effect on the absorption enhancement (e.g., Liu et al., 2015; Zhang et al., 2018; Xu et al., 2018; Wu et al., 2018). Indeed, we have shown that the SOA (LO-OOA + MO-OOA) were the main contributors to $E_{abs}$ during the cold period (Table 3 and Fig. 3).



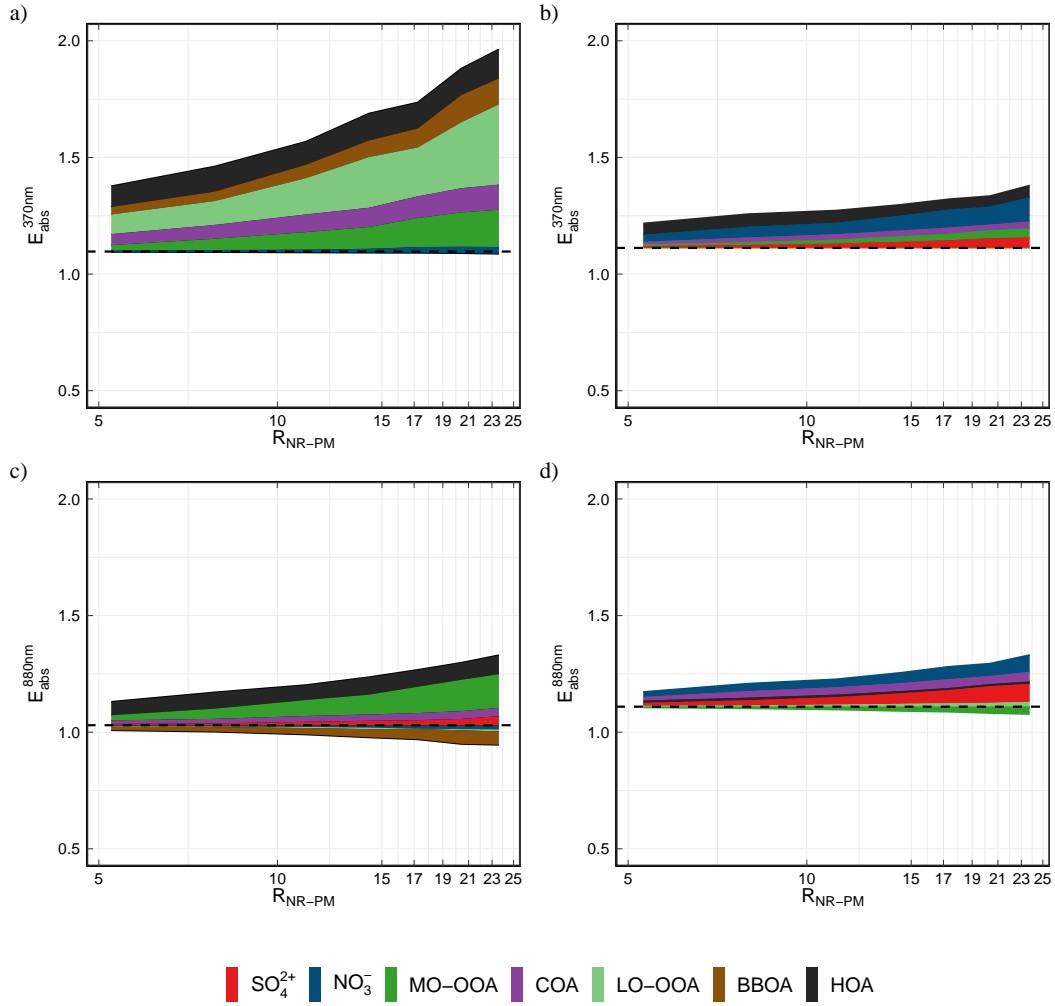

**Figure 3.** Contribution to the absorption enhancement, $E_{abs}$ as a function of the amount of coating material $R_{NR-PM}$ for the different organic and inorganic sources found at BCN during the cold period (left panel) and warm period (right panel) for 370 (upper panel) and 880 nm (lower panel). The contribution for each source was computed by applying the coefficient obtained with the mlr analysis (see Table 3) to the ratio that that compound-to-EC had as $R_{NR-PM}$ increased. It should be noted that for each season and wavelength, we have set the corresponding intercept of the mlr analysis as reference value above/below which each compound presented a positive/negative influence on $E_{abs}$.

Here we studied the behaviour of the absorption enhancement with the aging state of the aerosols. With this aim, we followed
the visual method proposed by Ng et al. (2010) to characterize the aging state of the particles, the so-called triangular plot, where the y-axis shows the $f_{44}$ factor (mz/44 to total concentration in the ACSM component mass spectra ratio), which is a proxy of the aging, and the x-axis is the $f_{43}$ factor (mz/43 to total concentration in the component mass spectra ratio), which





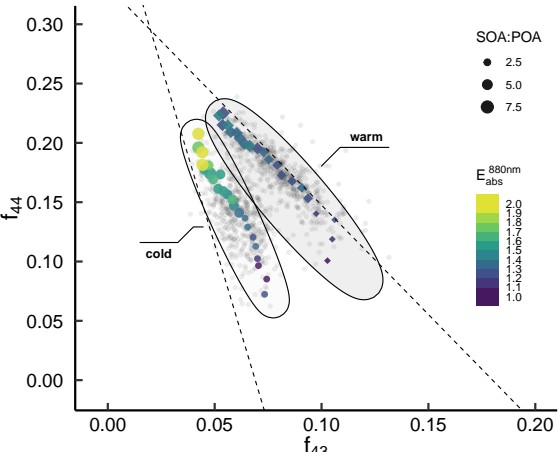

**Figure 4.** Absorption enhancement, $E_{abs}$, at 880 nm at BCN using online measurements as a function of the primary to secondary organic aerosol ratio (POA:SOA), and the atmospheric aging (following Ng et al. (2010) proposed triangle plot, $f_{44}$ vs $f_{43}$). The $f_{44}$ and $f_{43}$ factors used are the ones presented by Via et al. (2021) for the same time period from the Q-ACSM measurements.

shows the differences in the sources and chemical pathways for OOA formation. As particles become more oxidized they converge towards higher $f_{44}$ values and lower $f_{43}$ values.

As observed in Table 3 and Fig. 3 overall primary sources such as HOA, COA and BBOA were important drivers of the absorption enhancement, although SOA sources, especially during the cold period and for the shorter wavelengths, were also important sources contributing to $E_{abs}$. Figure 4 shows the $f_{44}$-$f_{43}$ relationship in BCN as a function of both $E_{abs}$ values at 880 nm, where $E_{abs}$ is driven by the lensing effect, and the SOA:POA ratio. Figure 4 shows a clear separation of the $f_{44}$-$f_{43}$ relationship between the cold and warm periods. In fact, during the cold period at 880 nm, as aerosols become more oxidized
(higher $f_{44}$), the SOA:POA ratio increased altogether with an increase of the absorption enhancement. Conversely, during the warm period, although the SOA:POA ratio also increased with the degree of oxidation, the absorption enhancement did not increase significantly. Therefore, during the cold period as particles became more oxidized, BC particles internally mixed with SOA (mainly MO-OOA as shown in Fig. 3c) were the main responsible for the larger $E_{abs}$ values, whereas during the warm period, since the main drivers of $E_{abs}$ were the inorganic compounds (Fig. 3d) higher $f_{44}$ values did not implied an
increase in $E_{abs}$. Furthermore, $E_{abs}$ at 370 nm (see Fig. S11) during the cold period showed a more pronounced increase as particles became more oxidized, mainly due to the role of externally mixed BrC, which, as reported in Fig. 3 were the main contributors to $E_{abs}$. Fig. S11 showed that $E_{abs}$ at 370 nm during the warmer period presented a slight increase as particles became oxidized, which could be attributed to the small contribution that MO-OOA particles exerted during this period (see Fig. 3b). This tendency toward higher $E_{abs}$ values as particles become more oxidized has also been found in Paris by Zhang
et al. (2018), and in London by Liu et al. (2015).





## 3.5 $E_{abs}$ trend analysis

As already stated, the average $E_{abs}$ values obtained by the offline method at BCN and MSY at 637 nm were within the values found in the literature for similar urban/regional background stations. Given the impact that the absorption enhancement of BC particles has on climate, we performed a seasonal trend analysis of $E_{abs}$ at both BCN (from 2011 to 2020), and MSY (from
2010 to 2020). In this trend analysis, as well as for the other results presented in this work, we excluded the days when Saharan dust outbreaks influenced the measurements to avoid UV absorption by dust in the analyses presented. Moreover, as already stated, the trend analysis was performed on $E_{abs}$ calculated at 637 nm, where the contribution to $E_{abs}$ from externally mixed OA was less relevant.

The method employed for the trend analysis was a Theil-Sen slope regression estimator. Previous studies performed at MSY
and BCN have shown statistically significant (s.s) decreasing trends over time for the contributions from various anthropogenic sources including traffic, industry, heavy-oil combustion, secondary sulphate and secondary nitrates mirroring the success of mitigation strategies adopted in Europe (Pandolfi et al., 2016; Veld et al., 2021). Moreover, recently (Veld et al., 2021) have shown that the observed decreasing trends, in combination with the absence of a trend for the organic aerosols (OA) at both BCN and MSY, resulted in an increase in the relative proportion of OA in PM at these stations, and especially for the SOA,
which presents the higher values during the summer season.

$E_{abs}$ trends showed different behaviours during spring-summer and autumn-winter periods at the two urban and background stations considered here (Fig. 5). Figure 5 shows a s.s. increase of $E_{abs}$ at MSY summer (JJA), whereas no s.s trends were observed at MSY during the other seasons. In BCN the $E_{abs}$ trends were no s.s. during all the seasons. During autumn (SON) and winter (DJF), $E_{abs}$ showed a slight decrease at both stations although not s.s. The $E_{abs}$ s.s. increase of a 8.16 % per
year during summer at MSY was linked to the observed increase of the OC:EC ratio (cf. Fig. S12), thus further confirming the importance of OA particles to form internal mixing with BC particles, thus increasing the $E_{abs}$. Conversely, the ratio sulphate:EC (Fig. S12 and S13) did not show any seasonal s.s. trend at both sites, mostly because both sulphates and EC concentrations decreased during the period under study (cf. Pandolfi et al., 2016; Veld et al., 2021). The observed OC:EC ratio increase at MSY in summer was mainly driven by the increase in the relative proportion of SOA particles as shown in
(Veld et al., 2021). We have shown here that as the aerosols become more oxidized the SOA:POA ratio increased together with $E_{abs}$ and the OC:EC s.s. trend observed further confirmed the importance of aged OA particles to form BC coating. In the case that the theoretical MAC was used as a reference (Bond and Bergstrom, 2006), the $E_{abs}$ showed the same behaviour with an increase at the regional station, MSY, during the summer months (Fig. S14).

During MAM of 2020 there was a notable $E_{abs}$ increase at both stations, and especially in BCN. During this period, a strict
lockdown was established in Spain due to restrictions under the COVID-19 pandemic. The strict lockdown measures implied a significant decrease in the emission of BC and primary aerosols due to the orders to halt any non-essential activity (Tobías et al., 2020; Evangeliou et al., 2021; Querol et al., 2021). This decrease in the primary emissions resulted in an increase in the OC:EC ratio, as can be appreciated in Fig. S12a, which can be associated with an increase in $E_{abs}$.





The observed changing behaviour of $E_{abs}$ under different SOA:POA ratios suggested that the absorption enhancement may

undertake changes, and possibly an increase, upon new emission restrictions. In fact, as already stated and as shown in Veld et al. (2021) an increase in the relative proportion of OA in $PM_{2.5}$ was observed at both BCN and MSY, and this relative increase was mostly due to SOA. Thus, based on our results, future increases of the SOA:POA ratio could cause an increase in $E_{abs}$.

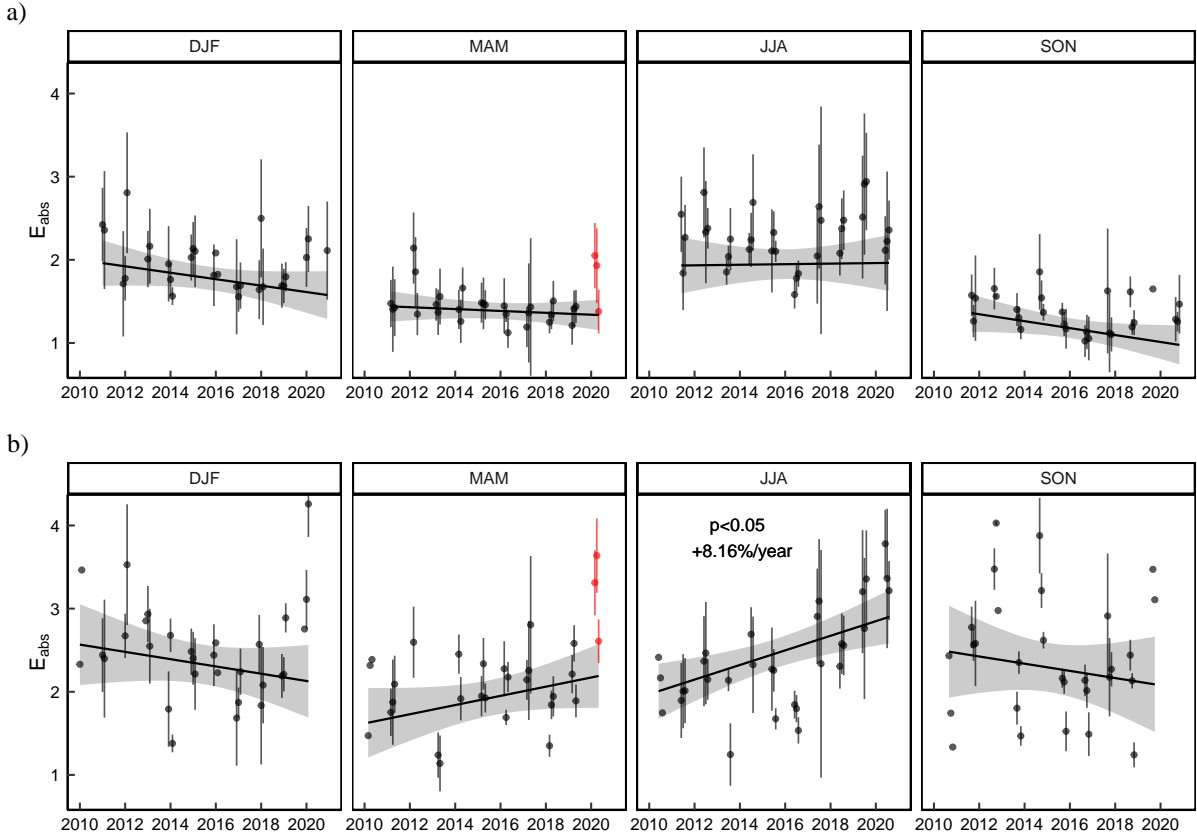

**Figure 5.** Absorption enhancement, $E_{abs}$ at 637 nm seasonal trend analysis between 2011 and 2020 at a) Barcelona and b) Montseny station. The trend analysis was performed using a Theil-Sen function over the $E_{abs}$ offline measurements.

## 4   Conclusions

Here we have presented the results of the analysis of absorption enhancement analysis, $E_{abs}$, performed in Barcelona (BCN, urban background) and Montseny (MSY, regional background) stations in the Mediterranean basin. We studied the main characteristics of $E_{abs}$ and its dependence on other chemical compounds using both an intensive online measurement period in





BCN (2018), and a decade-long offline dataset (2010-2020) available at both BCN and MSY. The online approach consisted of co-located measurements at BCN of multi-wavelength absorption coefficients with an aethalometer (AE33), OC:EC analysis through a Semi-Continuous Sunset Analyzer, and non-refractory fine aerosol speciation and source apportionment with a Q-ACSM. The offline method consisted in comparing MAAP absorption coefficient measurements (at 637 nm) with offline 24-hour offline OC:EC measurements performed via a thermal-optical carbon analyser, SUNSET, following the EUSAAR2 protocol.

We calculated $E_{abs}$ as the ratio between the ambient mass absorption cross-section (MAC) obtained from the measurements and the reference MAC value for pure BC particles. We have used two distinct reference MAC values: one based on an experimental site-specific MAC for pure BC, and a theoretical value from Bond and Bergstrom (2006). Using the site-specific reference MAC value, we reported $E_{abs}$ values of $1.28 \pm 0.36$, and $1.45 \pm 0.51$ for the online measurements at BCN at 880 nm and 370 nm, respectively, and of $1.42 \pm 0.40$ and $1.87 \pm 0.63$ for the offline analysis at BCN and MSY at 637 nm, respectively. The Eabs values reported in this work fall within the measured values reported in the literature (Liu et al., 2015; Zhang et al., 2018; Cappa et al., 2019). Moreover, our analysis confirmed the importance of OA particles as species that can increase $E_{abs}$ when these are both internally and externally mixed with BC particles, as also reported in Zhang et al. (2018) for the Paris area (France)

We showed here that the seasonal behaviour of $E_{abs}$ was a strong function of the wavelength used. In BCN we observed an increase of $E_{abs}$ at the near-ultraviolet wavelengths during the cold period and we related the observed increase to the presence of brown carbon (BrC) particles externally mixed with BC particles. Conversely, in the red and near-infrared spectral range the $E_{abs}$ variations were smaller. The relative contribution of BrC to the absorption enhancement increased from 4.6 % during the warm period up to 20.3% during the cold period, as expected due to the increase in the biomass burning activities during winter. $E_{abs}$ at MSY at 637 nm showed an increase during the warm period, mainly associated to the larger contribution of secondary organic aerosols (SOA) affecting the regional station due to the larger emission of biogenic precursors in summer.

In this study we performed an analysis on the influence that the amount of material available for BC coating exerted on $E_{abs}$. We showed, in agreement with some prior studies, an exponential growth of $E_{abs}$ with the amount of non-refractory aerosols. Thus, at the regional site, where the amount of material available for mixing reached higher values, so it did the $E_{abs}$ values. Moreover, when evaluating between the different wavelengths for the online measurements, we obtained higher values for the short-UV wavelength (370 nm), in comparison with the near-infrared wavelengths (880 nm), which was associated with the presence of externally mixed BrC increasing the absorption at the shorter wavelengths.

The aging state influence on $E_{abs}$ was examined using the triangular plot proposed by Ng et al. (2010) by means of the $f_{44}$ and $f_{43}$ factors derived from the Q-ACSM source analysis for online measurements at BCN station. We observed larger $E_{abs}$ values for more aged organic aerosols, especially during the cold period, which was also related with a larger ratio of secondary-to-primary organic aerosols.

The long database of offline filter and MAAP measurements at both BCN and MSY allowed for a decade long seasonal trend analysis of $E_{abs}$. Overall, no statistically significant trends were observed at both stations. The exception, however, was the summer period at MSY regional station where a statistically significant increasing trend of 8.16 % per year was



observed for $E_{abs}$. This increase of $E_{abs}$ at MSY in summer was mainly driven by a corresponding statistically significant increase of the OC:EC ratio. A previous study recently performed in the area under study, reported an increasing trend of the relative contribution of OA to PM and of SOA to OA with time at MSY regional station. Moreover, our analysis confirmed the importance of OA, and mostly of SOA, in contributing to the BC absorption enhancement. Furthermore, at both BCN and MSY the forced COVID-19 lockdown in spring 2020 implied a sharp increase of $E_{abs}$, mainly associated with the increase in the OC:EC ratio for this period due to the large reduction of anthropogenic emissions, and especially of BC particles, in the Barcelona urban environment. The observed statistically significant increasing trend of $E_{abs}$ at MSY in summer, driven by a corresponding increase in the OC:EC ratio, suggested that $E_{abs}$ could further increase during summer in the future due to the application of more restrictive measurements to reduce anthropogenic pollutant emissions. Thus, the higher absorption efficiency presented by the positive $E_{abs}$ trend offsets, to some extent, the reduction of the absorption that would be associated to the decreasing trend of BC particles concentrations.

*Code and data availability.* The Montseny data sets used for this publication are accessible online on the WDCA (World Data Centre for Aerosols) web page: http://ebas.nilu.no. The Barcelona data sets were collected within different national and regional projects and/or agreements and are available upon request. The code used for analysis can be obtained upon request to the corresponding author.

*Author contributions.* AA, MCM, AK, NP, CR, MP, MV, and JYD carried out the maintenance and supervision of the instrumentation at BCN and MSY sites. AA, XQ, MCM, and MP helped in the processes of shaping the manuscript structure as well as with the data analysis. MV and MCM performed the analysis of the Q-ACSM data, and AK the measurements of OC and EC both online and offline. JYD processed and merged the different data-sets, analyzed the results, and summarized and expressed them in this manuscript. All authors provided advice regarding the structure and content as well as contributed to the writing of the final manuscript.

*Competing interests.* At the time of the research, MR and MI were also employed by the manufacturer of the Aethalometer AE33.

*Acknowledgements.* Measurements at Spanish sites (Barcelona, Montseny) were supported by the Spanish Ministry of Economy, Industry and Competitiveness and I+D+I "Retos Colaboración" funds under the CAIAC project (PID2019-108990PB-100), by FEDER funds (EQC2018-004598-P), by the Generalitat de Catalunya (AGAUR 2017 SGR41 and the DGQA) and the European Commission via ACTRIS-IMP (project 871115). We acknowledge support of the COST Action CA16109 COLOSSAL. IDAEA-CSIC is a Centre of Excellence Severo Ochoa (Spanish Ministry of Science and Innovation, Project CEX2018-000794-S). We would like to thank the Parc Natural i Reserva de la Biosfera del Montseny and the Diputació de Barcelona for the possibility of maintaining the regional background measurement station.



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
