# Peer review of "Absorption enhancement of BC particles in a Mediterranean city and countryside: effect of PM chemistry, aging and trend analysis"

_Atmospheric Chemistry and Physics, 2022_

## Referee Comment (RC2)

The BC absorption enhancement due to the mixing is still unclear. This study reports the long-term measurements of $E_{abs}$ in Spain, which is valuable for better understanding the global distribution of BC absorption. Overall, it is well organized and professionally written. Therefore, I recommend this manuscript for publication after minor revisions.

In general, AE33 and MAAP are filter-based measurements. Several studies imply the corrections are needed for filter-based light absorption measurements, including multiple light scattering within the filter, filter loading, and particle scattering corrections (Lack et al., 2014; Moosmueller et al., 2009). Could you add the related description how the correction is down in this study? Moreover, OC/EC is a widely used instrument. But previous study shows there are several limitations associated with OC/EC measurement that complicate the interpretation of the results and introduce uncertainties that cannot be completely minimized (Lack et al., 2014). How do you think it affects your results?

It is very interesting to attribute $E_{abs}$ to different species. However, I wonder if the effects could be well estimated by using multiple linear regression due to the limitations of the method. Could you add discussion on the applicability of this method on this attribution?

Line 483: This study mentioned increase of $E_{abs}$ at the near-ultraviolet wavelengths during the cold period and we related the observed increase to the presence of brown carbon particles externally mixed with BC particles. Several studies estimate the impact of brown carbon internally mixed (brown carbon coating) with BC (Lack and Cappa, 2010; Feng et al., 2021). Is it different if brown carbon is internally mixed with BC particles?

Line 237: Table 2 occurs earlier than Table 1.

References

Feng, X., Wang, J., Teng, S., Xu, X., Zhu, B., Wang, J., Zhu, X., Yurkin, M. A., and Liu, C.: Can light absorption of black carbon still be enhanced by mixing with absorbing materials?, 253, 118358, https://doi.org/10.1016/j.atmosenv.2021.118358, 2021.

Lack, D. A. and Cappa, C. D.: Impact of brown and clear carbon on light absorption enhancement, single scatter albedo and absorption wavelength dependence of black carbon, 10, 4207–4220, https://doi.org/10.5194/acp-10-4207-2010, 2010.

Lack, D. A., Moosmueller, H., McMeeking, G. R., Chakrabarty, R. K., and Baumgardner, D.: Characterizing elemental, equivalent black, and refractory black carbon aerosol

particles: a review of techniques, their limitations and uncertainties, 406, 99–122, https://doi.org/10.1007/s00216-013-7402-3, 2014.

Moosmueller, H., Chakrabarty, R. K., and Arnott, W. P.: Aerosol light absorption and its measurement: A review, 110, 844–878, https://doi.org/10.1016/j.jqsrt.2009.02.035, 2009.

---

## Author Comment (AC1)

*Referee comment on "Absorption enhancement of BC particles in a Mediterranean city and countryside: effect of PM chemistry, aging and trend analysis" by Jesús Yus-Díez et al., Atmos. Chem. Phys. Discuss., https://doi.org/10.5194/acp-2022-145-RC1, 2022*

**Answer from the authors to Referee #1**

On behalf of all the authors of the manuscript, below we reply to the main comments presented by Referee #1. We would like to thank the Referee#1 for providing interesting and useful insights that we think contributed to improve the quality of this manuscript.

Hereafter we provide all the information and analysis required by the Referee#1.

All comments and/or changes we present below will be reported in the revised version of this manuscript.

**Comments.**

**1. The use of a constant correction factor ($C$) to account for the multiple scattering effect of AE33. According to the reference provided (AMT 2021, 14: 6335-6355), the $C$ values actually had considerable variations for the urban site (2.44 ± 0.57). Thus, instead of applying a constant $C$ to the AE33 results, the wavelength-resolved $b$abs should be determined using the MAAP-based $b$abs@637 nm and the AE33-based AAE, given that the $C$ values showed little wavelength dependence (AMT 2021, 14: 6335-6355).**

With this comment, the Referee#1 suggests to simulate the absorptions at the seven AE33 wavelengths using MAAP absorption data (at 637 nm) and the experimental AAE from AE33 absorption measurements.

As far as we know, the procedure suggested by the Referee#1 has never been reported in literature, thus we found this suggestion new and interesting. The basis for this suggestion is that the MAAP is generally considered as a "reference" instrument for absorption measurements because the filter-tape artefacts are dynamically calculated by the MAAP instrument. Conversely, AE33 data must be corrected off-line in order to consider the filter tape artefacts. Indeed, given that the C depends on the physical properties of the collected particles, the assumption of a constant C to correct the AE33 data leads to an overall higher uncertainty of the AE33 measurements compared to the MAAP measurements.

In fact, it has been reported (e.g. Zanatta et al., 2016; Rigler et al., 2020) that the absorption derived from AE33 data has a higher uncertainty (20-25%) compared to the MAAP (12%; Petzold and Schönlinner, 2004) and that this higher uncertainty is mostly due to the uncertainty associated to filter tape influence on the AE33 measurements. The standard deviation of the C reported in Yus-Diez et al. (2021) is around 23%. Yet the measurements reported in this manuscript have been calculated using error propagation laws and the effect of the measurement error in comparison with the standard deviation of the measurements is below 10%.

However, the data we presented in the manuscript prevent the application of the procedure suggested by the Referee#1 because MAAP data were collected with a cut-off inlet of PM10 whereas the AE33 measurements were performed in PM2.5. Thus, given that the semi continuous EC measurements used in this manuscript were also performed with a PM2.5 cut-off, the application of the procedure suggested by the Referee#1 could introduce an additional uncertainty

in the calculation of $E_{abs}$ due to the possible presence of coarse BC. In fact, as reported in Figure 1b below, on average offline EC concentrations in PM10 were 26% higher compared with online EC concentrations in PM2.5, whereas offline and online EC measurements in PM2.5 correlated well (slope = 1.02). A small bias (4%) between offline and online EC measurements in PM2.5 was also reported by Karanasiou et al. (2020).

[Figure]

**Figure 1:** Scatterplot between the offline 24-hour filter measurements of EC with an inlet cut-off of a) PM2.5 and b) PM10, and the online retrieved measurements of EC with an inlet cut-off of PM2.5.

Moreover, the figure below (Fig. 2) shows the relationship between the absorption at 660 nm measured with the AE33 and the absorption at the same wavelength extrapolated from MAAP measurements. As shown in the figure, there is a high correlation between the two absorptions with a slope of 1.1 (10% difference) and a positive intercept. Slope higher than 1 and positive intercept (even if small) were likely due to the different inlets used for AE33 and MAAP measurements.

[Figure]

**Figure 2:** Scatter plot between the MAAP absorption coefficients ($b_{abs,MAAP-660}$) extrapolated to 660 nm and the absorption coefficient at 660 nm from AE33 measurements, ($b_{abs,AE33-660}$).

Thus, with the data we have at disposal, we could only apply the procedure suggested by the Referee#1 using off-line EC measurements in PM10 from filter analysis to estimate $E_{abs}$ from MAAP-simulated absorptions. However, the filters were collected during 24h and only 2/3 filters per week were analyzed during the measurement period used for this manuscript. Thus, the application of the suggested procedure to filter data will dramatically reduce the temporal resolution and the amount of data available for this study.

Finally, we would also consider the fact that the AE33 instrument is the most widely used instrument worldwide for on-line attenuation measurements in monitoring stations. The MAAP instrument was discontinued a few years ago and only few stations nowadays deploy both AE33 and MAAP. At many monitoring stations only the AE33 is deployed and many papers have been published using the 7 absorptions obtained from AE33 measurements without MAAP data. Given the high correlation between the simulated and derived absorption reported in the Figure 2 above, we consider that the use of the seven absorptions provided experimentally by the AE33 is of higher interest for the scientific community and that the differences in $E_{abs}$ obtained from experimental and MAAP-simulated AE33 absorptions will be small compared to the total uncertainty.

The reasons causing the observed differences between online and offline measurements will be better commented in the revised version of the manuscript.

**2. Consistency of online and offline EC for the urban site. It is essential to present their relationship, e.g., using a scatter plot, and quantitatively determine the inter-method discrepancy. Unless this discrepancy could be properly accounted for, it does not make sense to compare the online and offline $E$abs (or MAC) results. In addition, the MAC of uncoated BC, i.e., MACref, were calculated for the urban site using both the online and offline EC. But it appears that the results differed substantially (Figures S1 and S2, after accounting for the wavelength dependence). This does not make sense, again raising concerns on whether the online and offline $E$abs (or MAC) results were comparable.**

Indeed, the inter-method discrepancies should have been better tackled in the submitted manuscript. Below, following the Referee#1 suggestion, we provide the necessary analysis to reply to this comment.

The differences between the online and offline MAC, $MAC_{ref}$ and $E_{abs}$ reported in the manuscript were most likely due to the different size cut-off used for online and offline measurements. As aforementioned, offline EC and absorption (from MAAP) measurements were performed in PM10 whereas the online EC and absorption (from AE33) measurements were performed in PM2.5.

As shown in Figure 1a above, PM2.5 offline and online EC measurements presented a very good correlation, in agreement with Karanasiou et al. (2020), and offline EC concentrations in PM10 were on average 26% higher compared to online EC concentrations in PM2.5 (Fig. 1b), and as shown in Fig. 2 above, the absorption at 660 nm in PM10 was on average 10% higher compared to the absorption measured in PM2.5. Besides the different inlets used, this difference was in part driven by the fact that different instruments (MAAP and AE33) were used to measure absorption.

However, the differences reported in Figs. 1 and 2 explained the differences reported in the manuscript for the MAC and $E_{abs}$.

Thus, we would like to highlight again that in our manuscript we used different techniques (and different cut-off) to estimate MAC and $E_{abs}$ and that this was the main reason for the differences between offline and online measurements reported in the manuscript. Despite this, we think that presenting results from different techniques in the manuscript is valuable because measurements with different cut-off and with different instruments (e.g. MAAP and AE33) are commonly performed worldwide. Moreover, we used the measurements we performed with different objectives rather than just for comparison between the techniques we used. In fact, online EC and AE33 measurements were used with the main objectives of studying the effect of chemistry and aging on the BC coating as a function of wavelength, and the offline EC and online MAAP measurements to study the trend of the MAC.

We would like to point out that the analysis presented here will be better clarified in the revised version of the manuscript, and that the corresponding figures will be included in the supplementary material.

**3. The assumption that "the lensing-driven absorption enhancement for BC particles was wavelength independent" was not supported by the references provided, even for the clear coating scenario (e.g., as indicated by Figure 5 in Lack and Cappa, ACP 2010).**
**4. The effects of brown coating were ignored, indicating the discussions on mixing state were highly uncertain, especially for the cold season when the influence of biomass burning was stronger.**

We would like to thank the Referee#1 for these comments. In fact, following the Referee#1 suggestions we changed the method to estimate the lensing driven BC absorption enhancement from experimental measurements. This new method presented below seems to confirm the wavelength dependence of the internally mixed BC absorption enhancement highlighted by the Refreee#1 and reported in other studies (mostly theoretical).
.
Below we reply to the comments 3 and 4 from Referee#1.

Indeed, we assumed that the absorption enhancement due to the internal mixing was wavelength independent, as stated in the manuscript lines 206-208: "Moreover, we assumed that the lensing-driven absorption enhancement for BC particles was wavelength independent (Lack and Cappa, 2010; Liu et al., 2015; Zhang et al., 2018)."

As correctly pointed out by Referee#1, we misinterpreted Lack and Cappa (2010). The main reason for our assumption was mostly based on the fact that the approximation of a wavelength independent $E_{abs}$ was used in some recent papers as for example in Liu et al. (2015) and Zhang et al. (2018). For example, Lack and Langridge (2013) commented (*author's note: $BC_{Int}$ corresponds to the attribution to the absorption of the internal mixing, i.e. coating, whilst $BC_{Ext}$ refers to the contribution from the pure BC particles*), that:

"As described previously, the theoretical AAE for $BC_{Int}$ can range from the uncoated baseline to ~ 1.7 (Gyawali et al., 2009; Lack and Cappa, 2010). In contrast, Bahadur et al. (2012) assumed that internal mixtures did not affect the AAE and used an AAE for $BC_{Int}$ = 0.55. Analysis of a range of atmospheric measurements of the AAE for aerosol sourced from fresh fossil fuel burning and urban pollution (where the dominant absorber was BC) shows an average value for the AAE of $1.1 \pm 0.3$ ($1\sigma$) derived using the wavelength pair 467 nm and 660 nm (Lack et al., 2008; Clarke et al., 2007; Virkkula et al., 2005; Rosen et al., 1978; Bergstrom et al., 2002, 2007; Kirchstetter et al., 2004). This suggests that the AAE extremes presented (0.55 and 1.7) are likely not common in the atmosphere for $BC_{Ext}$ and $BC_{Int}$, and serve here as extreme boundaries only. Although there is variability in the AAE, these studies have been used previously to support the use of an AAE = 1 for $BC_{Ext}$ (Bond et al., 2013), **and it is common to assume that the AAE for $BC_{Int}$ is equal to that of $BC_{Ext}$**. These studies provide evidence that although an AAE of 1 may be an accepted average for $BC_{Ext}$ and $BC_{Int}$, an uncertainty range should be considered and propagated through any absorption attribution procedure performed"

However, as correctly commented by the Referee#1, assuming a wavelength independent $E_{abs}$ is not supported by other studies (mostly theoretical studies) and by the Lack and Cappa (2010) reference provided in this manuscript. Indeed, Figs. 1 and 5 in Lack and Cappa (2010) show that under different core-shell and clear-brown coating scenarios, the absorption enhancement produced by the coating can vary with the wavelength.

In order to take into account this comment of the Referee#1 we recalculated the $E_{abs}(\lambda)$ following the procedure described below.

- First, differently from what was presented in the manuscript, we have used seven $MAC_{ref}$ values (calculated as the intercept of the relationship between the observed ambient MAC and the ratio OC:EC when the ratio is zero for each AE33 wavelength) to calculate $E_{abs}(\lambda)$ of coated BC, instead of using for all wavelengths the $E_{abs}$ value (1.20) for coated BC calculated at 880 as the ratio $MAC/MAC_{ref}$.
- Second, we performed a sensitivity study changing the AAE of coated BC, using the same AAE range as used in Fig. 1 from Lack & Langridge (2013) from 0.8 to 1.4.

Thus, in the revised version of the manuscript we calculated the contribution of the BC coating absorption to $E_{abs}(\lambda)$ as follows:

$$b_{abs}(\lambda) = b_{abs,BC\ core} + b_{abs,int} + b_{abs,BrC};$$

hence if we remove the contribution to the absorption by the BrC, then $E_{abs,BC\ coated}(\lambda)$ is:

$$E_{abs,BC\ coated}(\lambda) = \frac{b_{abs,BC\ core} + b_{abs,int}}{EC} \cdot \frac{1}{MAC_{ref}(\lambda)} =$$

$$= \frac{MAC_{BC\ coated}^{880\ nm} \cdot \left(\frac{880}{\lambda}\right)^{AAE} + MAC_{ref}(\lambda)}{MAC_{ref}(\lambda)} = 1 + \frac{MAC_{BC\ coated}^{880\ nm} \cdot \left(\frac{880}{\lambda}\right)^{AAE}}{MAC_{ref}(\lambda)}$$

Where, for the sensitivity study, different AAE were considered: 0.8, 1 and 1.4.

Thus, following Lack and Langridge (2013), we studied the variation of the absorption coefficients attributed to BC, the BC coating and BrC by varying the absorption Ångström exponent (AAE) of coated BC. An AAE below 1 has been observed for scenarios with large BC cores, whereas an AAE above 1 is often associated to the presence of brown coatings.

[Figure]

**Figure 3:** Contribution to the absorption coefficients of pure BC, coated BC, and BrC particles.

As it can be seen in Fig. 3, as the AAE increases/decreases above/below 1, the contribution to the absorption of the coating material increases/decreases with the wavelength. In fact, in the case of an AAE for coated BC of 1.4, the proportion of the absorption due to the BrC material becomes much smaller. An increase of the AAE of the internally mixed BC particles is linked with an increase in the relative contribution of brown material to the internal mixing (Zhang et al., 2020).

Indeed, Fig. 4 below shows that for AAE=1.4, the $E_{abs}$ for the internal mixing increases towards the shorter wavelengths, as observed in the simulations performed in Fig. 3 of Lack and Cappa (2010), where the dashed grey line represents the absorption enhancement produce by brown coating. In the case of AAE=1, the contribution of the coating material remains fairly constant (Fig. 4 below), although it presents a slight decrease with decreasing wavelengths, which is due

to the fact the MAC Ångström Exponent for the experimental reference MAC for pure BC particles is slightly above 1. Conversely, a clear decrease of the contribution to $E_{abs}$ from coated BC with decreasing wavelengths was observed for an AAE of 0.8, which is expected given that the contribution to the absorption from the coating decreases, as indicated by the AAE<1. Overall, this sensitivity analysis of the behavior of the impact of AAE of the coated BC particles shows that the assumption of a constant $E_{abs}$ for the internal mixing, although useful in a first approximation, is not always accurate. Thus, a variation in the AAE proves useful to determine the possible range of values of the $E_{abs}$ for the different mixing scenarios.

[Figure]

**Figure 4:** Absorption enhancement produced by the coating material and the BrC for different AAE values of the coating material.

However, since a modelization using Mie theory falls out of the scope of this work and given that modelling studies have been published that can be used as reference, we cannot determine how much the brown coating or the different possible core-shell particle diameters contribute to the observed behaviour of the absorption enhancement due to BC coating. We can only present, along with the sensitivity study, how much will vary the contribution of coated BC under different AAE scenarios for this coated BC, and by extension the BrC, to the absorption enhancement. Based on previous modelling studies, we could assume that, as reported in Fig. 5b of Lack and Cappa (2010), the observed contributions of coated BC to $E_{abs}$ could be related to a Bond et al. (2006) regime #2 with a BC particle central core diameter of 100 nm and a shell of 1500 nm.

The new analysis presented above will be included in the revised version of the manuscript.

**Bibliography**

1. Zanatta, Marco, et al. "A European aerosol phenomenology-5: Climatology of black carbon optical properties at 9 regional background sites across Europe." *Atmospheric environment* 145 (2016): 346-364.

2. Rigler, Martin, et al. "The new instrument using a TC–BC (total carbon–black carbon) method for the online measurement of carbonaceous aerosols." *Atmospheric Measurement Techniques*13.8 (2020): 4333-4351.

3. Petzold, Andreas, and Markus Schönlinner. "Multi-angle absorption photometry—a new method for the measurement of aerosol light absorption and atmospheric black carbon." *Journal of Aerosol Science* 35.4 (2004): 421-441.

4. Yus-Díez, Jesús, et al. "Determination of the multiple-scattering correction factor and its cross-sensitivity to scattering and wavelength dependence for different AE33 Aethalometer filter tapes: a multi-instrumental approach." *Atmospheric Measurement Techniques* 14.10 (2021): 6335-6355.

5. Karanasiou, Angeliki, et al. "Evaluation of the Semi-Continuous OCEC analyzer performance with the EUSAAR2 protocol." *Science of the Total Environment* 747 (2020): 141266.

6. Lack, D. A., and C. D. Cappa. "Impact of brown and clear carbon on light absorption enhancement, single scatter albedo and absorption wavelength dependence of black carbon." *Atmospheric Chemistry and Physics* 10.9 (2010): 4207-4220.

7. Liu, Shang, et al. "Enhanced light absorption by mixed source black and brown carbon particles in UK winter." *Nature communications* 6.1 (2015): 1-10.

8. Zhang, Yunjiang, et al. "Evidence of major secondary organic aerosol contribution to lensing effect black carbon absorption enhancement." *npj Climate and Atmospheric Science* 1.1 (2018): 1-8.

9. Lack, D. A., and J. M. Langridge. "On the attribution of black and brown carbon light absorption using the Ångström exponent." *Atmospheric Chemistry and Physics* 13.20 (2013): 10535-10543.

---

## Author Comment (AC2)

*Referee comment on "Absorption enhancement of BC particles in a Mediterranean city and countryside: effect of PM chemistry, aging and trend analysis" by Jesús Yus-Díez et al., Atmos. Chem. Phys. Discuss., https://doi.org/10.5194/acp-2022-145-RC21, 2022*

**Answer from the authors to Referee #2**

On behalf of all the authors of the manuscript, we would like to thank Referee #2 for the comments and suggestions to improve the manuscript. Below we provide all the information and analysis requested by the Referee#2.

All comments and/or changes we present below will be reported in the revised version of this manuscript.

**Comments.**

**In general, AE33 and MAAP are filter-based measurements. Several studies imply the corrections are needed for filter-based light absorption measurements, including multiple light scattering within the filter, filter loading, and particle scattering corrections (Lack et al., 2014; Moosmueller et al., 2009). Could you add the related description how the correction is down in this study? Moreover, OC/EC is a widely used instrument. But previous study shows there are several limitations associated with OC/EC measurement that complicate the interpretation of the results and introduce uncertainties that cannot be completely minimized (Lack et al., 2014). How do you think it affects your results?**

Indeed, filter-based measurements are characterized by numerous artifacts affecting the measurements. In relationship with the two instruments used in this study, AE33 and MAAP, the corrections performed were:

- AE33: Filter leakage was taken into account following the values provided by the manufacturer. The filter loading effect was corrected online by the instrument using the factor $k$, which corrects for the filter loading effect. In fact, the AE33 instrument uses the dual spot technology (Drinovech et al., 2015) that allows for an online correction for this artifact. Then, the multiple scattering parameter, C, was corrected using the value reported in Yus-Díez et al. (2021) for Barcelona station used in this study. The C was found to have an average value of 2.44, and did not present a marked dependence with the single scattering albedo (SSA) of the particles collected on the filter-tape. Yus-Díez et al. (2021) showed that the C values can considerably increase when SSA is high (> 0.95). However, these high SSA are rarely measured in the city of Barcelona (cf. Fig. 1). Moreover, Yus-Díez et al. (2021) reported that the C is wavelength independent in Barcelona (cf. Fig. 1). Therefore, we used the average value of 2.44 for the deriving the absorption measurements.

[Figure]

[Figure]

**Fig. 1:** The left panel represents the subplot c) of the Figure 1, and the right panel the subplot a) of Figure 4, respectively of Yus-Díez et al. (2021).

- MAAP: We followed Muller et al., (2011) recommendations for correcting the MAAP data, and we reported the absorption at 637 nm. The MAAP corrects online for filter tape artifacts by simultaneously measuring light transmitted through and scattered back from the particle laden filter. The only correction was considering the correction factor (1.05; Muller et al., 2011) due to the difference between the nominal (670 nm) and actual (637 nm) wavelength used by the instrument.

With regards to the limitations associated to the OC/EC measurements performed with the Sunset OC/EC analyzer, interferences in thermal-optical analysis of OC/EC in PM filter samples are well discussed in previous works. Different thermal protocols, light absorbing carbon, carbonates and other chemical components might influence the split point between EC and OC leading to overestimation or underestimation of EC concentrations (Kuhlbusch 2009; Karanasiou et al., 2015). For obtaining both the online and offline OC/EC measurements we used the reference methodology elaborated by WG35 of the European Committee for Standarization (CEN) that adopted the EUSAAR2 protocol with transmittance correction for OC/EC determination in PM2.5 (EN16909:2017). We calculated the combined relative standard uncertainty of EC concentrations using the method described in EN16909. This was equal to 18% for both online and offline OC/EC analyzers (Karanasiou et al., 2020).

**It is very interesting to attribute $E_{abs}$ to different species. However, I wonder if the effects could be well estimated by using multiple linear regression due to the limitations of the method. Could you add discussion on the applicability of this method on this attribution?**

Previous studies (e.g. Zhang et al., 2018) have used this technique to derive the contribution to $E_{abs}$ from different species. Figure 3 in the manuscript shows that the absorption enhancement depends on the concentration of particles available for mixing (non-refractory PM; $R_{NR-PM}$). Hence, by studying $E_{abs}$ variations with time and its relationship with the variations of each of the different sources contributing to $R_{NR-PM}$, we can infer the relative contribution of each source to the $E_{abs}$ variations. MLR analysis has been used in other papers with the objective of studying the relative importance of different sources/species to a given variable (e.g. Ealo et al., 2018; Zhang et al., 2020; among many others).

To check for the goodness of the fit, we performed a series of analysis, such as the VIF (Variance Inflation Factor), and the statistical significance. Overall, the tests showed good enough results that allow applying the MLR analysis.

This analysis can be find in a set of tables that were included in the Supplementary material of the revised version of the manuscript (Tables S2-S6 reported below):

**Table S2:** VIF (Variance Inflation Factor) between the independent variables of the multi-linear regression analysis, i.e. the chemical species and sources obtained with the Q-ACSM, and a test of the statistical significance using the p-value of each coefficient (*: $p<0.05$,**$<0.01$,***$<0.001$).

|  | Cold period | | Warm period | |
|---|---|---|---|---|
|  | VIF | p-value | VIF | p-value |
| **Intercept** | - | *** | - | *** |
| **HOA-to-EC** | 1.405 | * | 1.132 | * |
| **BBOA-to-EC** | 2.247 | ** | - | - |
| **MO.OOA-to-EC** | 6.045 | * | 3.015 | |
| **LO.OOA-to-EC** | 1.385 | * | 1.827 | |
| **SO4-to-EC** | 2.215 | | 1.913 | * |
| **NO3-to-EC** | 3.315 | | 1.207 | *** |
| **COA-to-EC** | 1.179 | ** | 1.515 | * |

As reported in Table S2, VIF values were in the acceptable range of values indicating that the independent variables were not correlated (VIF close to 1) or moderately correlated (VIF<5). The exception was for the MO-OOA/EC independent variable that showed a VIF of around 6 during the cold period. This means that the standard error for the regression coefficient of the MO-OOA/EC in winter was around 2.3 ($\sqrt{6.04}$) times larger than if that predictor variable had 0 correlation with the other predictor variables. However, in some studies (e.g. Vittinghoff et al., 2006; Hair, 2009) VIF < 10 has been considered as acceptable. Moreover, for the cold period the MLR analysis provided slightly negative values for SO4-to-EC and NO3-to-EC (Table 3 in the manuscript) indicating negligible contribution to $E_{abs}$ from these two variables during the cold period. In fact, the p-values in Table S2 for these two variables were not statistically significant (s.s.). However, the MO-OOA-to-EC ratio shows s.s. p-values indicating that the MLR analysis results are acceptable

**Line 483: This study mentioned increase of $E_{abs}$ at the near-ultraviolet wavelengths during the cold period and we related the observed increase to the presence of brown carbon particles externally mixed with BC particles. Several studies estimate the impact of brown carbon internally mixed (brown carbon coating) with BC (Lack and Cappa, 2010; Feng et al., 2021). Is it different if brown carbon is internally mixed with BC particles?**

We thank the Referee #2 for this comment. Indeed, with the method presented in the manuscript, by the assumption that the absorption enhancement due to internal mixing was constant, we could not account for the possible internal mixing of absorbing material (brown coating) with the BC cores. Lack and Cappa. (2010) have shown that in the case that there is an absorbing brown coating, actually, the absorption enhancement decreases towards the shorter wavelength as there is less light radiation reaching the BC cores (Figs. 1 and 5 in Lack and Cappa, 2010). Thus, BrC externally or internally mixed with BC can present different effects on the absorption enhancement.

To test this, as also suggested by the Refree #1, and to try to better incorporate the BrC internally mixed with the BC cores, we have recalculated the absorption enhancement for each wavelength, $E_{abs}(\lambda)$, following the procedure described below.

- First, differently from what was presented in the manuscript, we have used seven $MAC_{ref}$ values (calculated as the intercept of the relationship between the observed ambient MAC and the ratio OC:EC when the ratio is zero for each AE33 wavelength) to calculate $E_{abs}(\lambda)$ of coated BC, instead of using for all wavelengths the $E_{abs}$ value (1.20) for coated BC calculated at 880 as the ratio $MAC/MAC_{ref}$.
- Second, we performed a sensitivity study changing the AAE of coated BC, using the same AAE range as used in Fig. 1 from Lack & Langridge (2013) from 0.8 to 1.4.

Thus, in the revised version of the manuscript we calculated the contribution of the BC coating absorption to $E_{abs}(\lambda)$ as follows:

$$b_{abs}(\lambda) = b_{abs,BC\ core} + b_{abs,int} + b_{abs,BrC};$$

hence if we remove the contribution to the absorption by the BrC, then $E_{abs,BC\ coated}(\lambda)$ is:

$$E_{abs,BC\ coated}(\lambda) = \frac{b_{abs,BC\ core} + b_{abs,int}}{EC} \cdot \frac{1}{MAC_{ref}(\lambda)} =$$

$$= \frac{MAC_{BC\ coated}^{880\ nm} \cdot \left(\frac{880}{\lambda}\right)^{AAE} + MAC_{ref}(\lambda)}{MAC_{ref}(\lambda)} = 1 + \frac{MAC_{BC\ coated}^{880\ nm} \cdot \left(\frac{880}{\lambda}\right)^{AAE}}{MAC_{ref}(\lambda)}$$

Where, for the sensitivity study, different AAE were considered: 0.8, 1 and 1.4.

Thus, following Lack and Langridge (2013), we studied the variation of the absorption coefficients attributed to BC, the BC coating and BrC by varying the absorption Ångström exponent (AAE) of coated BC. An AAE below 1 has been observed for scenarios with large BC cores, whereas an AAE above 1 is often associated to the presence of brown coatings.

[Figure]

**Figure 3:** Contribution to the absorption coefficients of pure BC, coated BC, and BrC particles.

As it can be seen in Fig. 3, as the AAE increases/decreases above/below 1, the contribution to the absorption of the coating material increases/decreases with the wavelength. In fact, in the case of an AAE for coated BC of 1.4, the proportion of the absorption due to the BrC material becomes much smaller. An increase of the AAE of the internally mixed BC particles is linked with an increase in the relative contribution of brown material to the internal mixing (Zhang et al., 2020).

Indeed, Fig. 4 below shows that for AAE=1.4, the $E_{abs}$ for the internal mixing increases towards the shorter wavelengths, as observed in the simulations performed in Fig. 3 of Lack and Cappa (2010), where the dashed grey line represents the absorption enhancement produce by brown coating. In the case of AAE=1, the contribution of the coating material remains fairly constant (Fig. 4 below), although it presents a slight decrease with decreasing wavelengths, which is due

to the fact the MAC Ångström Exponent for the experimental reference MAC for pure BC particles is slightly above 1. Conversely, a clear decrease of the contribution to $E_{abs}$ from coated BC with decreasing wavelengths was observed for an AAE of 0.8, which is expected given that the contribution to the absorption from the coating decreases, as indicated by the AAE<1. Overall, this sensitivity analysis of the behavior of the impact of AAE of the coated BC particles shows that the assumption of a constant $E_{abs}$ for the internal mixing, although useful in a first approximation, is not always accurate. Thus, a variation in the AAE proves useful to determine the possible range of values of the $E_{abs}$ for the different mixing scenarios.

[Figure]

**Figure 4:** Absorption enhancement produced by the coating material and the BrC for different AAE values of the coating material.

However, since a modelization using Mie theory falls out of the scope of this work and given that modelling studies have been published that can be used as reference, we cannot determine how much the brown coating or the different possible core-shell particle diameters contribute to the observed behaviour of the absorption enhancement due to BC coating. We can only present, along with the sensitivity study, how much will vary the contribution of coated BC under different AAE scenarios for this coated BC, and by extension the BrC, to the absorption enhancement. Based on previous modelling studies, we could assume that, as reported in Fig. 5b of Lack and Cappa (2010), the observed contributions of coated BC to $E_{abs}$ could be related to a Bond et al. (2006) regime #2 with a BC particle central core diameter of 100 nm and a shell of 1500 nm.

The new analysis presented above has been included in the revised version of the manuscript in the methodology section 2.4 and the results section 3.1.1.

Text included in Sect. 2.4 (in bold), in lines 214-225:

"Furthermore, we have assumed here that BrC particles do not absorb at 880 nm (Kirchstetter et al., 2004) and that the measured absorption at this wavelength was only driven by the BC internally mixed particles (i.e. the lensing effect). **Moreover, although some studies assumed a wavelength independent lensing-driven absorption enhancement for BC particles (Liu et al., 2015; Zhang et al., 2018), other studies showed that the presence of brown coatings can produce variations in the spectral behaviour of $E_{abs}$ with the wavelength (Lack and Cappa, 2010). Consequently, in order to take into account the possible influences of the brown coatings on $E_{abs}$, following Lack and Langridge (2013) we performed a sensitivity study by studying the variation of the absorption enhancement attributed to BC, the BC coating and BrC by varying the absorption Ångström exponent (AAE) of internally mixed BC (cf. Fig. S5). For this, the absorption enhancement, $E_{abs}$ attributed to the different values of AAE for the internally mixed BC can be described as follows (Eq. 3):**

$$E_{abs,BC\ coated}(\lambda) = 1 + \frac{MAC_{BC\ coated}^{880\ nm} \cdot \left(\frac{880}{\lambda}\right)^{AAE}}{MAC_{ref}(\lambda)}, \tag{3}$$

**where for the sensitivity study presented here, different AAE (0.8, 1 and 1.4) were considered following Lack and Langridge (2013)."**

Text included in Sect. 3.1 (in bold), in lines 301-324:

"As already stated, ambient BC particles can be either externally or internally mixed with other aerosols (Bond and Bergstrom, 2006). In order to separate the relative contributions to $E_{abs}$ of these two mixing states, i.e. external ($E_{abs,ext}$) and internal ($E_{abs,int}$) we used the multi-wavelength AE33 and the semi-continuous OC:EC measurements obtained in BCN (see Sect. 2.4). We assumed that the $E_{abs}$ at the near-infrared (880 nm) was only produced by the internal mixing of BC particles, whereas at the short-UV (370 nm) the Eabs is due to both the internal and external mixing of BC particles. Given the spectral characteristic of BrC absorption, the contribution to $E_{abs}$ due to external mixing was the highest at 370 nm compared to the other AE33 wavelengths. **In addition, here we analyzed the possible contribution of different internal mixing states of BC using different AAE for internally mixed BC, since the presence of brown coatings over the BC cores can actually produce a reduction of the enhancement of the absorption towards the shorter wavelengths (cf. Lack and Cappa, 2010).**

**Figure 2 shows the evolution of the contribution of the internal and the external mixing to the total $E_{abs}$ for the three AAE values considered for internally mixed BC. Indeed, Fig. 2 shows that an AAE of 0.8 could be related with a larger proportion of brown coatings reducing the absorption enhancement due to the internally mixed BC (cf. Fig. 5 Lack and Cappa, 2010). In the case of AAE=1, the contribution of the coating material remains fairly constant (Fig. 2), although it presents a slight decrease with decreasing wavelengths, which is due to the fact the MAC Ångström Exponent for the experimental reference MAC for pure BC particles is slightly above 1 (Fig. S6). Moreover, Fig. 2 shows that for an AAE of 1.4 the internal mixing increases towards the shorter wavelengths, as observed in the simulations performed in Fig. 3 of Lack and Cappa (2010) for the case of BC core with a brown shell that does not absorb.**

**The overall contribution due to the internal mixing ($E_{abs,int}$) ranged between a 100% at 880 nm, and 83, 86, and 93.5% of the total $E_{abs}$ at 370 for an AAE of 0.8, 1 and 1.4, respectively. Thus, the BrC externally mixed particles represented a non-negligible fraction of the total $E_{abs}$ at near-ultraviolet wavelengths (Table S1), especially for the AAE=0.8 case, for which it increased from 0.069 ± 0.066 (5.2%) at 660 nm up to 0.17 ± 0.18 at 370 nm (16.9%). Conversely, if an AAE=1.4 is used, then the increase and relative contribution of $E_{abs}$ due to the BrC externally mixed particles remains lower, from 0.023 ± 0.049 (1.7%) at 660 nm up to 0.093 ± 0.200 at 370 nm (6.5%).**"

**Line 237: Table 2 occurs earlier than Table 1.**

We thank the referee for taking notice. Line 237, now 252 has been changed to: "in Table 1 and Table 2, respectively."

**Bibliography**

1. Petzold, Andreas, and Markus Schönlinner. "Multi-angle absorption photometry—a new method for the measurement of aerosol light absorption and atmospheric black carbon." *Journal of Aerosol Science* 35.4 (2004): 421-441.
2. Müller, T. *et al.* Characterization and intercomparison of aerosol absorption photometers: Result of two intercomparison workshops. *Atmos. Meas. Tech.* 4, 245–268 (2011).
3. Yus-Díez, Jesús, et al. "Determination of the multiple-scattering correction factor and its cross-sensitivity to scattering and wavelength dependence for different AE33 Aethalometer filter tapes: a multi-instrumental approach." *Atmospheric Measurement Techniques* 14.10 (2021): 6335-6355.

4. Zhang, Yunjiang, et al. "Evidence of major secondary organic aerosol contribution to lensing effect black carbon absorption enhancement." *npj Climate and Atmospheric Science* 1.1 (2018): 1-8.

5. Ealo, Marina, et al. "Impact of aerosol particle sources on optical properties in urban, regional and remote areas in the north-western Mediterranean." *Atmospheric Chemistry and Physics*18.2 (2018): 1149-1169.

6. Zhang, Yunjiang, et al. "Substantial brown carbon emissions from wintertime residential wood burning over France." *Science of the Total Environment* 743 (2020): 140752.

7. Vittinghoff, Eric, et al. "Regression methods in biostatistics: linear, logistic, survival, and repeated measures models." (2006): 139-202.

8. Hair, Joseph F. "Multivariate data analysis." (2009).

9. Lack, D. A., and C. D. Cappa. "Impact of brown and clear carbon on light absorption enhancement, single scatter albedo and absorption wavelength dependence of black carbon." *Atmospheric Chemistry and Physics* 10.9 (2010): 4207-4220.

10. Liu, Shang, et al. "Enhanced light absorption by mixed source black and brown carbon particles in UK winter." *Nature communications* 6.1 (2015): 1-10.

11. Zhang, Yunjiang, et al. "Evidence of major secondary organic aerosol contribution to lensing effect black carbon absorption enhancement." *npj Climate and Atmospheric Science* 1.1 (2018): 1-8.

12. Lack, D. A., and J. M. Langridge. "On the attribution of black and brown carbon light absorption using the Ångström exponent." *Atmospheric Chemistry and Physics* 13.20 (2013): 10535-10543.

13. Zhang, Yunjiang, et al. "Evidence of major secondary organic aerosol contribution to lensing effect black carbon absorption enhancement." *npj Climate and Atmospheric Science* 1.1 (2018): 1-8.

---

## Author Response (AR2)

**Answer from the authors to referee #1 comments on the revised version of the manuscript:** *"Absorption enhancement of BC particles in a Mediterranean city and countryside: effect of PM chemistry, aging and trend analysis" by Jesús Yus-Díez et al., Atmos. Chem. Phys. Discuss.*

Hereafter we will answer and resolve the comments by Referee #1.

1. **Please clarify the cut-off sizes of MAAP and AE33 which were used to derive the C factor.**

   The cut-off sizes used for deriving the C factor were the same as the ones used through-out this study: a PM2.5 inlet cut-off for the AE33 and a PM10 inlet cut-off for the MAAP. This experimental configuration was used to determine and characterize the C correction factor in Yus-Díez et al. (2021) where we assumed that most of the BC is contained in the PM2.5 fraction.

   We have clarified this in the manuscripts in lines 143-149:

   "*For the AE33, the larger uncertainty is introduced by the multiple scattering parameter, C ($\delta C = \pm 0.57$ at BCN Yus-Díez et al., 2021), which depends on the physical properties of the particles collected on the filter tape. In Yus-Díez et al. (2021) the C, **obtained with the same instruments (i.e. MAAP and AE33) and inlets cut-off as in the present work**, was found to have an average value of 2.44, and it did not present a marked dependence with the single scattering albedo (SSA) of the particles collected on the filter-tape. In fact, Yus-Díez et al. (2021) showed that the C values can considerably increase when SSA is high (> 0.95). However, these high SSA are rarely measured in the city of Barcelona. Moreover, it was reported that the C is wavelength independent in Barcelona (cf. Fig. 1 Yus-Díez et al., 2021). Therefore, we used here the average C value of 2.44 for the deriving the absorption measurements.*"

2. **Clarify the cut-off sizes for the on-line and off-line results, and when necessary, clearly state that the on-line and off-line results could not be directly compared, e.g., for the MAC values in Table 1.**

   We have modified the manuscript so that this is stated more clearly. Hereafter we provide a list of the lines where we have included this comment.

   In the methodology section, in lines 139-142:

   "MAAP measurements were obtained with a 1 min time resolution at a flow rate of 5 l/min and with a **PM10 inlet cut-off**. The AE33 $b_{abs}$ coefficients in BCN were derived with the same time resolution and flow rate as the MAAP and with **a PM2.5 inlet cut-off**. The aethalometer filter loading effect was corrected online by the dual-spot manufacturer correction (Drinovec et al., 2015), and the multiple scattering correction parameter, C, was set to 2.44, as obtained for the station by Yus-Díez et al. (2021)."

   With regards to the results section, in lines 260-263:

   "The difference between the offline and online measurements at BCN, although the mean values fall within the standard deviation of the measurements, was mainly associated to the difference in the length of the measurement periods, and especially the **different inlet cut-offs sizes, which prevents direct comparison** (Fig. S1)."

Additionally, Table 1 caption has been modified to:

"**Table 1**. Observed MAC ($m2g-1$) values obtained using online techniques via AE33 and Sunset online EC measurements at BCN for a PM2.5 inlet cut-off, and offline at BCN and MSY via MAAP and offline EC measurements on 24-hour filters for a PM10 inlet cut-off."

In Section 3.2, when comparing both the online-offline methods, it is stated the importance of the difference in the inlet cut-off size in lines 355-357:

"The higher $E_{abs}$ in BCN at 637 nm compared to $E_{abs}$ at 370 nm was mostly associated to the different inlets size cut-offs and, to a lesser extent, to the different periods used for the online and offline measurements."

And in lines 362-366:

"These different trends between online and offline $E_{abs}$ versus RNR−PM were probably due to two main factors: first, the offline measurements were made with a PM10 inlet vs the PM2.5 inlet of the online method (Fig. S1), hence coarse nitrates and other coarse particles could have influenced $E_{abs}$, and, second, the large annual variability observed for the offline $E_{abs}$ measurements (see Fig. S12) could have also contributed to the observed difference."